



# Improving NOX emissions in Beijing using network observations and a novel perturbed emissions ensemble

Le Yuan[1], Olalekan A.M. Popoola[1], Christina Hood[2], David Carruthers[2], Roderic L. Jones[1], Haitong Zhe Sun[1], Huan Liu[3], Qiang Zhang[4], and Alexander T. Archibald[1,5]

[1]Yusuf Hamied Department of Chemistry, University of Cambridge, Cambridge, CB2 1EW, UK.
[2]Cambridge Environmental Research Consultants, Cambridge, CB2 1SJ, UK.
[3]State Key Joint Laboratory of ESPC, State Environmental Protection Key Laboratory of Sources and Control of Air Pollution Complex, International Joint Laboratory on Low Carbon Clean Energy Innovation, School of the Environment, Tsinghua University, Beijing, 100084, China.
[4]Ministry of Education Key Laboratory for Earth System Modeling, Department of Earth System Science, Tsinghua University, Beijing, 100084, China.
[5]National Centre for Atmospheric Science, Cambridge, CB2 1EW, UK.

*Correspondence to*: Alexander T. Archibald (ata27@cam.ac.uk)

**Abstract.** Emissions inventories are crucial inputs to air quality simulations and represent a major source of uncertainty. Various methods have been adopted to optimise emissions inventories, yet in most cases the methods were only applied to total anthropogenic emissions. We have developed a new approach that updates *a priori* emission estimates by source sector, which are particularly relevant for policy interventions. At its core is a perturbed emissions ensemble (PEE), constructed by perturbing parameters in an *a priori* emissions inventory within their respective uncertainty ranges. This PEE is then input to an air quality model to generate an ensemble of forward simulations. By comparing the simulation outputs with observations from a dense network, the initial uncertainty ranges are constrained and *a posteriori* emission estimates are derived. Using this approach, we were able to derive the transport sector NOX emissions for a study area centred around Beijing in 2016 based on *a priori* emission estimates for 2013. The absolute emissions were found to be 1.5-9×10$^4$ Mg, corresponding to a 57-93% reduction from the 2013 levels, yet the night-time fraction of the emissions was 67-178% higher. These results provide robust and independent evidence of the trends of traffic emission in the study area between 2013 and 2016 reported by previous studies. We also highlighted the impacts of the chemical mechanisms in the underlying model on the emission estimates derived, which is often neglected in emission optimisation studies. This work paves forward the route for rapid analysis and update of emissions inventories using air quality models and routine *in situ* observations, underscoring the utility of dense observational networks. It also highlights some gaps in the current distribution of monitoring sites in Beijing which result in an underrepresentation of large point sources of NOX.



## 1 Introduction

Nitrogen dioxide ($NO_2$) is an important atmospheric trace gas whose adverse health impacts have been extensively studied. Controlled human exposure experiments have shown associations between short-term exposure to very high levels of $NO_2$ and airway inflammation (Blomberg et al., 1999), increased bronchial reactivity (Folinsbee, 1992), increased susceptibility to respiratory virus infections (Goings et al., 1989), etc. Chronic exposure to lower doses of $NO_2$ (e.g. those currently

observed in Europe and North America) has also been linked to lower lung function and deficits in lung function growth among children (Gauderman et al., 2000; Peters et al., 1999), chronic respiratory symptoms (Zemp et al., 1999) and increased cardiopulmonary mortality (Hoek et al., 2002) among adults in epidemiological studies. A key challenge for these epidemiological studies is to separate out the health effects due to $NO_2$ exposure from those due to exposure to other pollutants, whose concentrations are often highly correlated with those of $NO_2$.


$NO_2$ belongs to the highly reactive group of nitrogen oxides ($NO_X$), whose emissions occur primarily in the form of nitric oxide (NO) with a small proportion of $NO_2$ (i.e. $NO_X = NO + NO_2$). NO is quickly oxidised by ozone ($O_3$) to $NO_2$ which, in daylight hours, rapidly photolyses to reform NO and (via $O(^3P)$) $O_3$. Thus, during daytime, NO, $NO_2$ and $O_3$ reach a photostationary state, typically on the time scale of a few minutes (Leighton, 1961). The presence of volatile organic

compounds (VOCs) perturbs this null cycle by producing organic peroxy radicals ($RO_2$) which oxidise NO without consuming $O_3$, leading to faster NO to $NO_2$ conversion and net $O_3$ production. This results in a non-linear response of $O_3$ concentrations to reductions in the emissions of $NO_X$ and VOCs (Seinfeld and Pandis, 2016). It is therefore crucial to have accurate emission estimates for developing effective and synergetic control strategies for these interdependent pollutants (Cohan et al., 2005).


Though $NO_X$ can be produced from natural sources such as microbial processes and lightning, they are predominantly released from anthropogenic sources including fossil fuel combustion and open biomass burning (Feng et al., 2020; Lee et al., 1997). Global total anthropogenic $NO_X$ emissions flattened around 2008, as reductions in Europe and North America were offset by increases in Asia (Hoesly et al., 2018). China, in particular, witnessed a rapid rise in $NO_X$ emissions until 2011-

2012 (with the exceptions of a few regions where the emissions peaked earlier), which resulted from economic growth along with an absence of regulations (van der A et al., 2017; Liu et al., 2016; Zheng et al., 2018). Emission reduction targets were first announced in the 12[th] Five-Year Plan (2011-2015), followed by the Action Plan on Prevention and Control of Air Pollution (2013-2017) and the Three-Year Action Plan for Winning the Blue Sky Defence Battle (2018-2020). The main measures implemented included the installation of selective catalytic reduction equipment in power generating and industrial

facilities and the implementation of stricter vehicle emission standards combined with accelerated fleet turnover (Liu et al., 2020).



Numerous studies have quantified China's NO$_X$ emissions and evaluated the short- or long-term trends in emissions. Some have used a bottom-up method which combines energy consumption data from individual source sectors with the corresponding emission factors (Liu et al., 2016; Zhang et al., 2009; Zhao et al., 2013; Zheng et al., 2018). The substantial amount of data required not only results in an inevitable time-lag between the occurrence of emissions and the establishment of an inventory (Janssens-Maenhout et al., 2015), it also propagates potentially large and poorly quantified uncertainties into the emission estimates (Hong et al., 2017; Zhao et al., 2011). Other studies have inferred top-down estimates of emissions using satellite observations due to their continuous spatiotemporal coverage and near-real time availability (Ding et al., 2020; Lin et al., 2010; Qu et al., 2017; Zhang et al., 2012). This method requires tropospheric column densities of NO$_2$, which are retrieved by transforming slant column densities to vertical column densities, removing the stratospheric contribution, and correcting for the effects of albedo, cloud and aerosol effects (Leue et al., 2001). Earlier studies used a mass balance approach that assumes a linear relationship between emission rates and column densities (Martin et al., 2003) or between the normalised differences in the two quantities (Lamsal et al., 2011). The linear coefficient was determined from a chemical transport model (CTM) using an *a priori* emissions inventory and could then be used to derive *a posteriori* emission estimates from satellite retrievals of column quantities. The linear relationship in one grid cell is assumed to be unaffected by atmospheric transport and chemistry in neighbouring grid cells (Mijling and Van Der A, 2012; Streets et al., 2013). For pollutants of longer lifetimes or at finer model resolutions, however, it is important to account for non-local sensitivities of pollutant concentrations to emissions. Advanced data assimilation techniques such as Kalman Filter (Napelenok et al., 2008), ensemble Kalman Filter (Miyazaki et al., 2012) and four-dimensional variational assimilation (Kurokawa et al., 2009) have been increasingly adopted to combine satellite observations and CTM simulations. These inverse methods are time consuming and computationally demanding. The *a posteriori* emission estimates are also subject to uncertainties propagated from the satellite retrievals or the model simulations. For instance, Archer-Nicholls et al. (2021) showed large differences in the NO$_2$ column density simulated with two chemical mechanisms integrated into the same model with identical NO$_X$ emissions, which resulted from different treatment of non-methane volatile organic compounds (NMVOCs) and thus the conversion of NO$_X$ to sinks and reservoir species (via reactions with oxidation products of NMVOCs).

This study is aimed at optimising *a priori* NO$_X$ emissions in a bottom-up inventory compiled for Beijing for the year 2013 to account for emissions in 2016 using a novel approach. Uncertainties associated with emission trends between 2013 and 2016 were sampled by a perturbed emissions ensemble (PEE), which was constructed on the basis of an expert elicitation. The PEE was then input to an atmospheric dispersion model to generate an ensemble of air quality simulations. By comparing the simulated surface concentrations of NO, NO$_2$ and O$_3$ with observations from a dense monitoring network, the initially estimated uncertainties could be reduced, and *a posteriori* emissions could be derived. The sensitivity of the results to the chemical mechanisms in the model was also evaluated.



## 2 Methods

### 2.1 Observations

Emission estimates were constrained using pollutant concentrations measured in ground-based networks of high spatiotemporal resolution. $NO_2$ and $O_3$ are measured hourly at the long-term air quality monitoring sites operated by the Beijing Municipal Environmental Monitoring Center using reference instruments. Figure 1 shows the 33 sites that were in operation in 2016 and located within the study area (also see Table S1), determined by extent of the base emissions (see Sect. 2.2), and a classification according to the local environment. Traffic monitoring sites are situated up to 20 m from the kerbside of major roads, while urban and suburban sites monitor air quality in built-up areas not in close proximity to traffic in the six central districts and the outer districts, respectively. Clean and regional background sites that are away from built-up areas and major pollution sources measure the baseline concentrations. In addition, measurements at the regional background sites are representative of pollution transport from and to neighbouring regions (Ministry of Environmental Protection of the People's Republic of China, 2013b). The $NO_2$ and $O_3$ analysers are required to have a measurement range of 0-500 ppb with a minimum detection limit $\leq 2$ ppb and precision $\leq 10$ ppb when measured with span gas at 80% of the full span (Ministry of Environmental Protection of the People's Republic of China, 2013a). We used provisional real-time measurements from 2016 archived at https://quotsoft.net/air/ (last accessed 22 August 2020), as ratified historical data are not publicly available.

In addition, we used high frequency (20 s) measurements of NO, $NO_2$ and $O_3$ from November-December 2016, the winter campaign period of the Atmospheric Pollution & Human Health in a Chinese Megacity (APHH-Beijing) research programme (Shi et al., 2019). The measurements were made with low-cost sensors also deployed in a variety of settings in Beijing and are hereinafter referred to as SNAQ (Sensor Network for Air Quality) (Fig. 1 and Table S2). NO, $NO_2$ and $O_3$ were measured by electrochemical sensors with detection limit values of $< 4$ ppb, $< 1$ ppb and $< 1$ ppb, respectively (Mead et al., 2013; Popoola et al., 2018). The dataset has been validated against reference instrument measurements also obtained during the campaign and those from the aforementioned long-term monitoring sites.


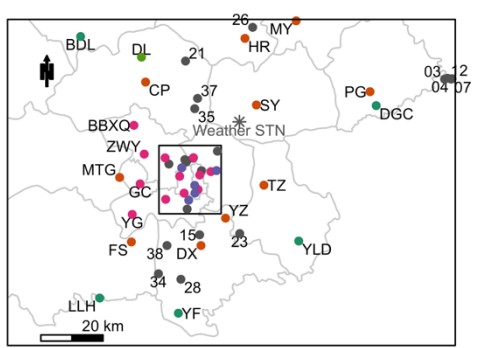

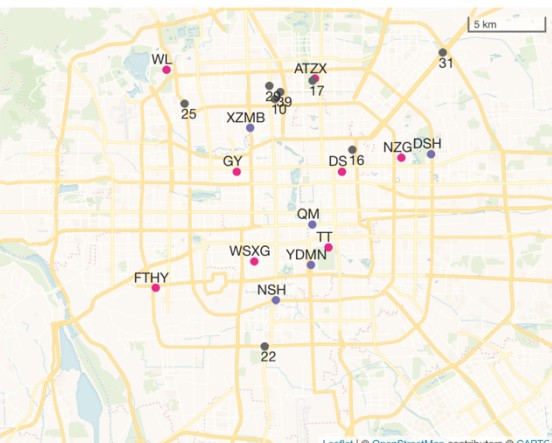

**Figure 1. The modelling domain as set by the extent of the base emissions (see Sect. 2.2) with the locations of air quality measurements used in this study, including a magnified view of the area within the 5th Ring Road of Beijing (right panel). Long-term monitoring sites are colour-coded according to the site type and labelled by their acronyms. Full names and coordinates are**
125 **listed in Table S1. Locations of low-cost sensor (SNAQ) measurements are shown in dark grey, and the coordinates can be found in Table S2. The weather station where the input meteorological observations were made is marked by the grey star symbol. The administrative divisions of Beijing are shown by light grey outlines.**

## 2.2 Perturbed emissions ensemble

130   We used a special version of the Multi-resolution Emission Inventory for China (MEIC) v1.3 (Li et al., 2017; Zheng et al., 2018) developed for use in the APHH-Beijing programme. The inventory characterises emissions of CO, $NO_X$ (and $NO_2$), total VOC (TVOC), $SO_2$, $PM_{10}$ and $PM_{2.5}$ from the industry, power, residential and transport sectors in 2013. It extends 120 km and 150 km in the North-South and East-West directions, respectively, covering most of Beijing and parts of Hebei Province with 3 km × 3 km horizontal resolution. In the vertical, there are seven layers with the top of each layer at 38, 90,

135   152, 228, 337, 480 and 660 m above ground, respectively. Emissions from each source sector are associated with distinct diurnal, monthly and vertical variation profiles that apply to all pollutants. This inventory has been used to simulate street level air quality (Biggart et al., 2020) and quantify regional pollution transport (Panagi et al., 2020) and has been compared with direct flux measurements (Squires et al., 2020). To focus on locations where observations were available, we cropped the original extent to a smaller region of 105 km × 144 km (starting from the Northwest) and used it as the *a priori*

140   emissions, hereinafter referred to as the base emissions.



NO$_X$ and TVOC emissions in Beijing were reported to have decreased substantially between 2013 (the year of the emission estimates) and 2016, when the observations were made (Cheng et al., 2019; Xue et al., 2020). In the surrounding provinces, NO$_X$ emissions also revealed a downward trend, while no apparent trend has been identified for TVOC emissions (Zheng et al., 2018). In addition to spatial disparities, emissions from individual source sectors also showed different patterns. In Beijing, for example, vehicle emission control contributed the most NO$_X$ reductions, while the largest TVOC decrease was found in the petrochemical industry (Cheng et al., 2019; Xue et al., 2020). Hence, uncertainties associated with the NO$_X$ emissions from each sector in the base emissions were estimated separately. Due to a lack of long-term observations, uncertainties associated with the TVOC emissions were not investigated, the impact of which on the constrained NO$_X$ emissions is discussed in Sect. 4.

To reduce subjectivity, the uncertainties were determined based on elicitation of expert knowledge. Table 1 shows the NO$_X$ emission parameters investigated. To facilitate the expert elicitation and the subsequent construction of PEEs, the parameters were defined as ratios of the 2016 values to the corresponding 2013 estimates in the base emissions. An exception is the last parameter which represents the night-time fraction of transport sector NO$_X$ emissions in 2016, irrespective of that in 2013. It allowed for perturbations to the diurnal distributions of traffic NO$_X$ emissions on top of perturbations to the total magnitude. In the initial PEE (see below and Sect. S1), the night-time fraction was defined as emissions occurring between 11 pm and 6 am (inclusive) following Biggart et al. (2020), who provided evidence of an underestimation in the night-time vehicle sources of NO$_X$ in the *a priori* emissions inventory. This was attributed to an underrepresentation of emissions from heavy duty diesel trucks, which typically travel from surrounding provinces into Beijing at night, as they are banned from entering the central urban areas during the day. After reviewing a previous study which summarised the varying traffic rules and restrictions for different types of vehicles in Beijing (Zhang et al., 2019), the definition was modified to traffic NO$_X$ emitted during 0 – 5 am (inclusive) for the optimised PEE.

To simultaneously perturb the total magnitude and the vertical distribution of emissions, the three-dimensional industry sector was split into two parameters, namely ground-level emissions (i.e. from the lowest vertical layer) and elevated emissions (i.e. from all upper layers). This was also intended for the power sector. However, as emissions from the sector are present in all but the lowest layer, their vertical variation profile was effectively unchanged in the initial PEE. The issue was fixed by introducing two new parameters for power sources below and above 152 m (top height of the 4[th] vertical layer), respectively, for the optimised PEE. Residential and transport emissions are only found in the ground layer and were thus represented each by a single parameter.

Those who participated in the elicitation included researchers with expertise in compiling emissions inventory for the region of interest and researchers who used the same *a priori* emissions inventory in their own work. For each emission parameter, they were invited to advise a lower and an upper bound of uncertainty, such that it would be very unlikely for the true value





to fall outside this range. The responses from the first round of elicitation were sent back to the participants anonymously for review. Finally, the maximum and minimum values advised by all participants for each parameter in the second round were adopted (Table 1, column *Initial PEE*).

Model simulations using the initial PEE as inputs showed substantial overestimation of $NO_2$ concentrations, such that many members of the ensemble were unusable for constraining the emissions (see Sect. S1). Hence, we designed an optimised PEE by decreasing the elicited lower bounds of uncertainty for all parameters concerning the magnitude of $NO_X$ emissions from a certain source sector. The upper bound of uncertainty for the transport emissions was also reduced, as the modelled diurnal concentration profiles indicated positive biases linked specifically to the sector. Lastly, the uncertainty range of the

night-time fraction of transport emissions was adjusted following the new definition described above (Table 1, column *Optimised PEE*).

**Table 1. Emission parameters[a] and the respective uncertainty ranges sampled by the initial and the optimised perturbed emissions ensembles (PEEs).**


| Parameter | Initial PEE | | Optimised PEE | |
|---|---|---|---|---|
| | Min | Max | Min | Max |
| Industry sector ground level $NO_X$ emissions | 0.4 | 1.6 | 0.05 | 1.6 |
| Industry sector elevated $NO_X$ emissions | 0.4 | 1.4 | 0.05 | 1.4 |
| Power sector $NO_X$ emissions[b] | 0.2 | 1.4 | NA | NA |
| Power sector $NO_X$ emissions below 152 m | NA | NA | 0.05 | 1.6 |
| Power sector $NO_X$ emissions above 152 m | NA | NA | 0.05 | 1.6 |
| Residential sector $NO_X$ emissions | 0.4 | 1.5 | 0.05 | 1.5 |
| Transport sector $NO_X$ emissions | 0.4 | 2 | 0.05 | 1.5 |
| Night-time fraction of transport sector $NO_X$ emissions | 0.1 | 0.4 | 0.1 | 0.3 |

[a] The night-time fraction of transport sector $NO_X$ emissions is defined as a proportion (%) of the daily totals in 2016. Other parameters are defined as ratios of the 2016 emissions to the base emissions from 2013.

[b] Power sector $NO_X$ emissions are effectively represented by one parameter in the initial PEE. In the optimised PEE, the

emissions are split into two parameters, namely emissions below and above 152 m.

Additional parameters were defined for CO emissions, the uncertainty ranges of which were also elicited and modified. The constrained emission estimates have been presented in Yuan et al. (2021). As discussed in that paper, CO is treated as an inert pollutant in the model used (see Sect. 2.3), thus varying CO emissions do not affect the modelled $NO_X$ concentrations



(and vice versa). This justified a simultaneous perturbation of all parameters. The 14 parameters in total (i.e. 7 for $NO_X$, 7 for CO) determined for the optimised PEE constituted a 14-dimensional uncertain space, which was probed efficiently using the maximin Latin hypercube sampling, which maximises the minimum inter-sample distance (Johnson et al., 1990). A rule of thumb is to have a sample size 10 times the dimension (Loeppky et al., 2009). We drew 140 samples, effectively doubling the sample size generally required (i.e. if only $NO_X$ emission parameters were perturbed). The sample values were then used

as spatiotemporally uniform scaling factors to perturb the corresponding values in the base emissions to construct a 140-member PEE, hereinafter referred to as the optimised PEE. Figure 2 shows the total $NO_X$ emissions by source sector and vertical layer and the mean diurnal variations of $NO_X$ emissions in the ensemble members. In each member, the set of scaling factors applied to $NO_X$ was also applied to the emissions of $NO_2$, such that primary $NO_2$ (f-$NO_2$, i.e. the proportion of NOx emitted directly as $NO_2$) in the base emissions remained unchanged, with a value of 6.7% in all source sectors (and thus

grid cells). In reality, however, the f-$NO_2$ varies between sectors. Much attention has been paid to the f-$NO_2$ in vehicle exhausts, while little is known about the f-$NO_2$ in residential emissions. It is thus difficult to evaluate whether the 6.7% in the base emissions is representative of the aggregated $NO_X$ emissions in the study area.





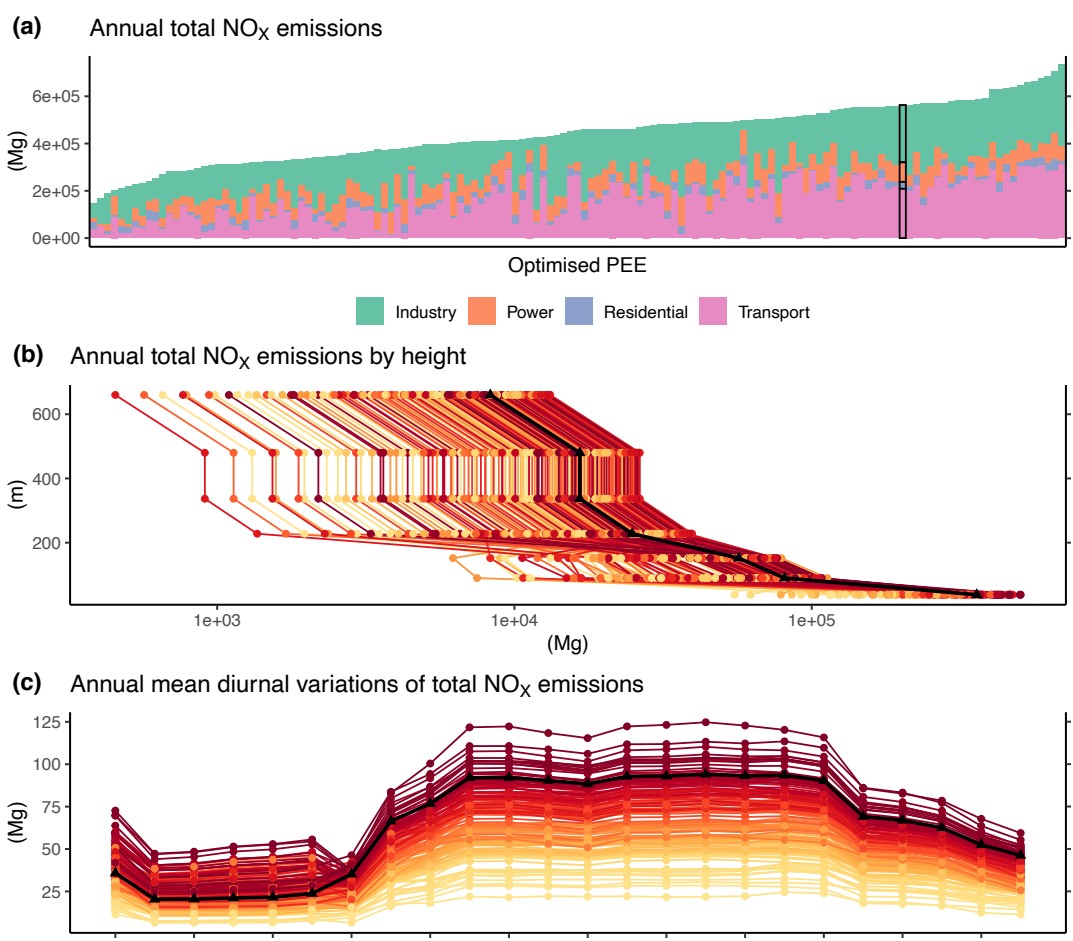

**Figure 2. (a)** Annual total NO$_X$ emissions shown by contributions from individual source sectors in the 140-member optimised perturbed emissions ensemble (PEE) and the base emissions (marked by black frames). **(b)** Vertical distributions of the annual total NO$_X$ emissions in the optimised PEE and the base emissions. The height represents the top height (above local ground level) of each vertical layer. **(c)** Annual mean diurnal variations (in local time) in total NO$_X$ emissions in the optimised PEE and the base emissions. In panels **(b)** and **(c)**, the base emissions are marked by black lines and triangle symbols, while the optimised PEE members are colour-coded according to their annual total NO$_X$ emissions with darker colour indicating higher values.

## 2.3 Model description and simulation setup

We used ADMS-Urban (version 4.2), a state-of-the-art urban scale high resolution quasi-Gaussian dispersion model (McHugh et al., 1997; Owen et al., 2000) to simulate pollutant concentrations. The model has been applied in air quality simulations in cities worldwide including Beijing (Biggart et al., 2020; He et al., 2019; Hood et al., 2018). Dispersion calculations are based on the state of the atmospheric boundary layer which is parameterised based on Monin-Obukhov similarity theory (Venkatram, 1996). The parameterisation is explained in detail in previous studies (e.g. Biggart et al., 2020).



The minimum required meteorological input data including hourly wind speed, wind direction and cloud cover were measured at a weather station at the Beijing Capital International Airport (see Fig. 1) and archived in the NOAA Integrated

Surface Database (Smith et al., 2011). To account for differences in local near-surface dynamics between the weather observatory (situated in open landscape) and the measurement sites (the majority of which are located in built-up areas), different roughness lengths and minimum Obukhov lengths were defined (Yuan et al., 2021).

Chemistry calculations are enabled by two fast chemistry schemes for the $NO_X$ photolytic chemistry and the formation of

sulphate aerosols, respectively (Cambridge Environmental Research Consults Limited, 2017). The former is based on the Generic Reaction Set (Azzi et al., 1992) which reduces the complex mechanisms involving $NO_X$, $O_3$ and VOCs to seven reactions:

$$ROC + h\nu \rightarrow RP + ROC \tag{R1}$$

$$RP + NO \rightarrow NO_2 \tag{R2}$$

$$NO_2 + h\nu \rightarrow NO + O_3 \tag{R3}$$

$$NO + O_3 \rightarrow NO_2 \tag{R4}$$

$$RP + RP \rightarrow RP \tag{R5}$$

$$RP + NO_2 \rightarrow SGN \tag{R6}$$

$$RP + NO_2 \rightarrow SNGN \tag{R7}$$

$$2NO + O_2 \rightarrow 2NO_2 \tag{R8}$$

where ROC, RP, SGN and SNGN represent reactive organic compounds, radical pool, stable gaseous nitrogen product and stable non-gaseous nitrogen product, respectively. Reaction (R8) has been added to the scheme in ADMS-Urban, but its impact is only significant with sustained high levels of NO concentrations (e.g. 1000 μg m$^{-3}$ for several hours) due to a small rate constant (Cambridge Environmental Research Consults Limited, 2017). It is evident that only reactions (R3) and (R4)

are conservative chemical reactions, while the rest represent approximations of multiple reactions lumped together. For example, reaction (R1) represents all reactions that produce radicals via the photo-oxidation of VOCs. Thus, the rate coefficients of these generic reactions have been determined empirically by fitting the simulation outputs to smog chamber data (Azzi et al., 1992). The rate constant of the explicit reaction (R3) can be calculated from solar radiation, which is often estimated by ADMS-Urban based on the input meteorological data, when direct measurements are unavailable (as is the case

in this study). By appealing to $NO_X$-$O_3$ photostationary state the model also derives a $NO_2$ photolysis rate from the background concentrations of NO, $NO_2$ and $O_3$ and takes the lower value between the two (Cambridge Environmental Research Consults Limited, 2017).





Background pollutant concentrations are thus required as an input, not only to account for pollution sources not included in
the input emissions (e.g. transported from outside the extent of the emissions inventory), but also to constrain the reaction
coefficients for reactive species. As mentioned in Sect. 2.1, continuous measurements of $NO_2$ and $O_3$ are available from the
long-term monitoring network. In each hour, we input the inverse distance weighted mean of the concentrations at two of the
clean or regional background sites (a total of six, see Fig. 1) located to each side of the incoming wind direction in that hour
as the background in the optimised PEE simulations. This was different from the initial PEE simulations (see Sect. S1)
which used a baseline concentration, defined as the $10^{th}$ percentile of the concentrations from all sites in a moving 3-h
window. The method of using a network baseline to represent the non-local pollution signal has previously been applied to
CO, which is inert in ADMS-Urban (Yuan et al., 2021), and $NO_X$ and $O_3$ at a small spatial scale (Popoola et al., 2018). At
larger scales, however, though it has been established that concentrations of total oxidants ($O_X = O_3 + NO_2$) consist of a
local component that correlates with $NO_X$ emissions and a $NO_X$-independent regional component (Clapp and Jenkin, 2001;
Han et al., 2011), the partition of $O_X$ between $NO_2$ and $O_3$ may be highly variable in time and space due to their rapid
interconversion and reactions with other species. Using values measured at two neighbouring sites away from major sources
ensured a more realistic partition of $O_X$ in the optimised PEE simulations. The sensitivity of the simulation output to
different definitions of $NO_2$ and $O_3$ background concentrations is further investigated in Sect. 4.

For NO and TVOC for which long-term measurements were unavailable, we used upwind concentrations from the
Copernicus Atmosphere Monitoring Service (CAMS) reanalysis dataset (Inness et al., 2019). At each time step (every 3
hours), we calculated the inverse distance weighted mean of the values from the two grid cells in the lowest vertical layer
located directly outside of the modelling domain and to each side of the incoming wind. The time series obtained was then
linearly interpolated to hourly resolution, as required by ADMS-Urban. As TVOC is not a standard output variable in the
dataset, a sum of the 8 available VOC species was used to approximate TVOC. To validate this approach, we compared the
sum of mixing ratios of 29 VOC species measured at the Institute of Atmospheric Physics, Chinese Academy of Sciences
during the APHH-Beijing winter campaign with the approximate TVOC mixing ratios from the corresponding grid cell in
the reanalysis product during the same period. Apart from a few peak events not seen in the latter, the two time series show a
good level of agreement both in terms of the trend and the magnitude (Fig. 3a). The upwind NO mixing ratios extracted from
the CAMS reanalysis dataset were compared to NO baseline mixing ratios extracted from the SNAQ measurements. Figure
3b shows a substantial positive bias in the NO extracted from the reanalysis dataset, the cause of which remains unknown.
To prevent this bias from being propagated into the modelled concentrations, a bias correction was applied using empirical
quantile mapping. This method equates the (empirically estimated) cumulative distribution functions (i.e. quantile functions)
of the modelled and observed time series for regularly spaced quantiles (Boé et al., 2007; Cannon et al., 2015). During the
campaign period, this significantly reduced the bias while the correlation was only slightly decreased. However, larger
uncertainties would have been introduced when the transfer function was extrapolated to the entire time series of 2016,





which were unavoidable and difficult to quantify due to a lack of long-term measurements. Yet these were likely smaller than the uncertainties associated with using the uncorrected, positively biased values obtained from the CAMS reanalysis product.


**(a)** TVOC at IAP

**(b)** Background NO

**Figure 3. (a)** Mixing ratios of TVOC at the Institute of Atmospheric Physics (IAP), Chinese Academy of Sciences during the APHH-Beijing winter campaign from the Copernicus Atmosphere Monitoring Service (CAMS) reanalysis dataset, compared to observations. TVOC from the reanalysis product was approximated by the sum of 8 available VOC species. The observed TVOC was calculated as the sum of 29 VOC species measured. **(b)** Original and bias-corrected upwind mixing ratios of NO from the reanalysis dataset (the latter were input as background pollution levels in the PEE simulations), compared to baseline (10th percentile) mixing ratios from the SNAQ measurements. For each reanalysis time series, the data and the normalised mean bias (NMB) and Pearson's correlation coefficient (r) are shown in the same colour.





The input meteorology data and background pollutant concentrations described above provided the same lateral boundary
conditions for all optimised PEE simulations, among which only the emissions of $NO_X$ (and $NO_2$) varied. An additional
simulation forced with these boundary conditions and the base emissions was also performed and is hereinafter referred to as
the base run.

## 3 Results

We first evaluated the performance of the optimised PEE simulations in modelling $NO_2$ concentrations at the long-term
monitoring sites using mean square error (MSE). It is a compact indicator of model performance whose merit is
demonstrated in the following. The MSE is calculated as:

$$MSE = \frac{1}{n}\sum_{i=1}^{n}(mod_i - obs_i)^2 \,, \tag{1}$$

where $obs_i$ is the observed value for a given averaging period $i$ (e.g. an hour, a day, a month), $mod_i$ is the corresponding
simulation output and $n$ represents the length of the available observations.

At each site, MSE is calculated for the hourly $NO_2$ concentrations output by each optimised PEE simulation and the base run.
Figure 4a shows a distinct trend of increasing MSEs with growing annual total $NO_X$ emissions, such that at most sites, the
base run with input emissions at the upper end of the scale (see Fig. 2a) is outperformed by most of the PEE simulations.
Although MSE does not differentiate between over- and underestimation, this clear positive association suggests a positive
bias in the base emissions. It is also evident that the base run is generally associated with larger errors at urban and traffic
monitoring sites compared to other sites. This is also seen in most individual PEE simulations. Moreover, the errors
associated with the ensemble of simulations typically span a wider range at these locations. This is an indication that the
magnitude of emissions in the central areas (where these sites are situated, see Fig. 1) are higher than those in the periphery
and overestimated to a larger extent in the base emissions. Though spatially uniform scaling factors were applied within the
study area, regions with a higher value in the base emissions would show larger variations (as a result of being larger in
magnitude). This highlights a potential issue associated with spatially uniform perturbations to spatially non-uniform
emissions.

Due to the important role of $O_3$ in converting NO into $NO_2$, the optimised PEE simulations' performance in modelling the $O_3$
concentrations was also evaluated. In other words, this was to ensure that the underlying chemical mechanisms were
correctly modelled and that simulations with lower $NO_X$ emissions showed better agreement with $NO_2$ observations for the
right reasons. We calculated the MSEs in maximum daily 8-hour mean (MDA8) $O_3$ concentrations[1]. Among the numerous

---

[1] The optimised PEE simulations' performance in modelling the hourly $O_3$ concentrations was also evaluated. The median MSEs in hourly
$O_3$ concentrations of all simulations are dominated by the mMSE, and their association with the input $NO_X$ emissions is substantially





O₃ metric available, the MDA8 O₃ is widely used for model-observation comparison due to its relevance in regulation and

health impact assessments (Lefohn et al., 2018). Figure 4b shows that as with NO₂, the base run is also generally associated with higher MSEs than many PEE simulations. At more sites, however, the positive association between model error and NOₓ emissions seen in Fig. 4a breaks down (e.g. at DSH and MTG) or even becomes reversed (e.g. at HR and LLH). This underscores the complex effects of non-linear chemistry and suggests that the MSEs in MDA8 O₃ are less strongly associated with the input NOₓ emissions, the reason for which can be revealed by a breakdown of the MSE.


weaker than the association between median MSEs in MDA8 O₃ concentrations and NOₓ emissions.





**(a)**   MSE in hourly NO₂ concentrations

$(\mu g\ m^{-3})^2$  ▇ [597, 1549)  ▇ [1549, 2285)  ▇ [2285, 3983)  ▇ [3983, 13399]

**(b)**   MSE in MDA8 O₃ concentrations

Increasing NO$_X$ emissions →

$(\mu g\ m^{-3})^2$  ▇ [310, 538)  ▇ [538, 728)  ▇ [728, 1334)  ▇ [1334, 6656]

**Figure 4. Mean square errors (MSE) in (a) hourly NO₂ concentrations and (b) daily maximum 8-hour mean (MDA8) O₃ concentrations associated with the optimised perturbed emissions ensemble (PEE) simulations, arranged in ascending order of the input annual total NO$_X$ emissions (from left to right), and the base run (marked by black frames) at each long-term monitoring site. In each panel, the MSEs are grouped into quartiles and colour-coded accordingly. The monitoring sites are colour-coded**





according to the site type: **urban site (magenta), traffic monitoring site (purple), suburban site (orange), clean site (light green) and regional background site (green).**

The MSE can be mathematically decomposed into the sum of three terms (Solazzo and Galmarini, 2016):

$$\text{MSE} = \ [(\overline{\text{mod}} - \overline{\text{obs}})^2] + [(\sigma_{mod} - r\sigma_{obs})^2] + [\sigma_{obs}^2 \times (1 - r^2)] \qquad (2)$$

where $\sigma_{mod}$ and $\sigma_{obs}$ represent the standard deviation of the modelled and observed values, respectively, and $r$ is the Pearson's correlation coefficient between model outputs and observations. The first term in Eq. (2) represents the bias component of model error and it is largely introduced by external forcings, for example, input emissions and boundary conditions. The second term is the variance error which is associated with the processes resolved in a model. A trade-off

between bias and variance, in other words, accuracy and precision, is often inevitable in complex models (Sun and Archibald, 2021). The last term, by definition, represents the proportion of the observed variance unexplained by the model. It summarises all non-systematic errors such as noise and inherent variability (e.g. due to turbulence closure) in the observations and is referred to as the minimum achievable MSE (mMSE). The MSE is thus a well-rounded metric suitable for operational model evaluation, and its decomposition provides indications of possible sources of model error (Solazzo and

Galmarini, 2016).

The values of MSEs shown in Fig. 4 were decomposed according to Eq. (2) and the term with the largest contribution to the MSE associated with each simulation at each site is shown in Fig. 5. There are striking differences in the attribution of error for hourly $NO_2$ and MDA8 $O_3$. Variance errors have the largest share in the MSEs in hourly $NO_2$ concentrations associated

with most of the optimised PEE simulations at over half of the reference sites. At another 1/4 of the sites, all of which are urban or traffic monitoring sites, bias accounts for most of the errors in the bulk of the simulations. This is another indication of a higher degree of overestimation in the central areas in the base emissions and subsequently in many members of the optimised PEE. For most sites (regardless of the site type), simulations using the lowest annual total $NO_X$ emissions are associated with MSEs that are mostly made up by the mMSE.


At the urban and traffic monitoring sites where the MSEs in hourly $NO_2$ associated with the ensemble simulations are mainly made up of the bias error, the bias also happens to be the largest term in most MSEs in MDA8 $O_3$ (Fig. 5b). At most other sites (including all suburban, clean and regional background sites), however, the MSEs in MDA8 $O_3$ are dominated by the mMSE, irrespective of the input $NO_X$ emissions. This also explains the weaker association between the total MSEs in MDA8

$O_3$ and the emissions, as the mMSE is much less dependent on model inputs (see Fig. 4b). A further breakdown of the mMSEs (Fig. S3) reveals that variances in the observations of MDA8 $O_3$ are substantially higher than those in the observed hourly $NO_2$ concentrations. Despite considerably better correlation between model outputs and the observations, these large variances in the observations result in mMSEs in MDA8 $O_3$ concentrations that are only moderately smaller than those associated with hourly $NO_2$ concentrations. Meanwhile, the MSEs in $O_3$ are considerably lower than those in $NO_2$, the





largest share of the mMSEs in the former is thus explained. Because of this dependence on the observations and the weaker connection to external drivers, the mMSE is often considered the least concerning component of model error (Solazzo and Galmarini, 2016).





**(a)**  Decomposed MSE in hourly NO$_2$ concentrations

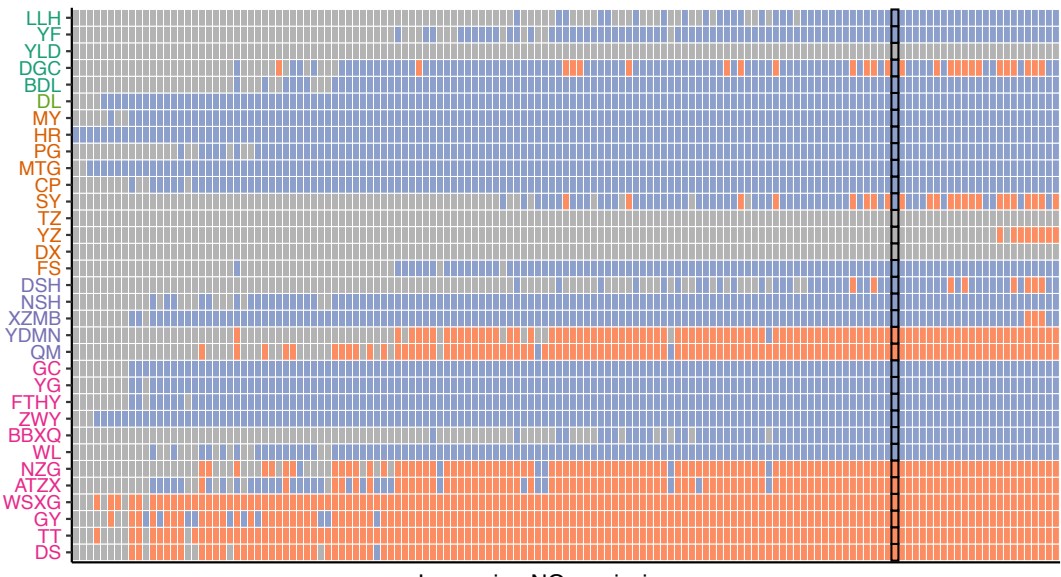

Increasing NO$_X$ emissions →

**(b)**  Decomposed MSE in MDA8 O$_3$ concentrations

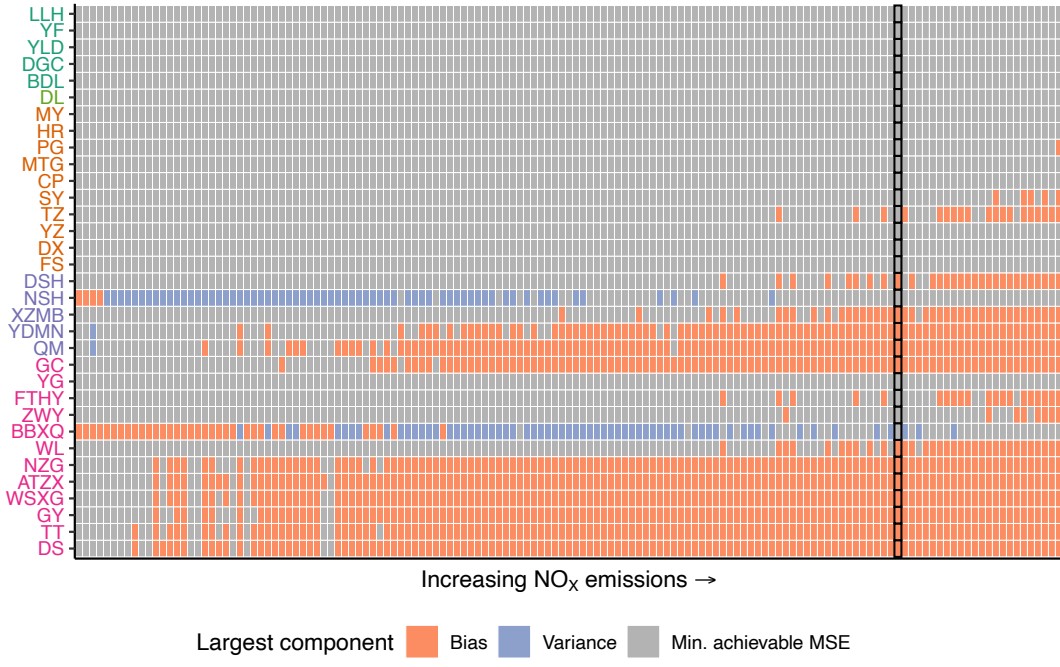

Increasing NO$_X$ emissions →

Largest component  ▮ Bias  ▮ Variance  ▮ Min. achievable MSE

**Figure 5. Error component with the highest contribution to the mean square errors (MSE) in (a) hourly NO$_2$ concentrations and (b) daily maximum 8-hour mean (MDA8) O$_3$ concentrations associated with the optimised perturbed emissions ensemble (PEE) simulations, arranged in ascending order of the input annual total NO$_X$ emissions (from left to right), and the base run (marked by black frames) at each long-term monitoring site. The monitoring sites are colour-coded according to the site type: urban site (magenta), traffic monitoring site (purple), suburban site (orange), clean site (light green) and regional background site (green).**




As the distributions of the 33 MSEs (i.e. one for each long-term monitoring site) associated with individual PEE simulations are mostly non-Gaussian, we used the median MSE to represent a simulation's average performance for a certain pollutant across all sites within the modelling domain. A breakdown of the median MSEs (Fig. 6) is consistent with the findings described above. With more accurate (lower) input $NO_X$ emissions, a simulation's average performance for hourly $NO_2$ can be improved substantially to a point that the remaining model error consists mostly of the non-systematic mMSE. The average performance for MDA8 $O_3$ is less strongly associated with $NO_X$ emissions, as the model error is dominated by the mMSE in the majority of the PEE simulations. These associations between median MSEs and input emissions are also tested using simple linear regression (Fig. S4). Though both regression models are statistically significant (p-value < 0.001), more variability in the modelled hourly $NO_2$ is explained, compared to that in the modelled MDA8 $O_3$. The scatter of the points around the regression line reveals the variations in model performance with varying mix of source sectors, given similar strengths of total emissions (note that these are represented on an ordinal scale in Fig. 4 – Fig. 6). This also demonstrates the importance of using network observations as constraints. Similar concentrations at a particular location may result from several different combinations of emission parameter values. The risk of constraining the parameter values to just one of the possible combinations is reduced, when observations that sample a wide range of local environments are used.




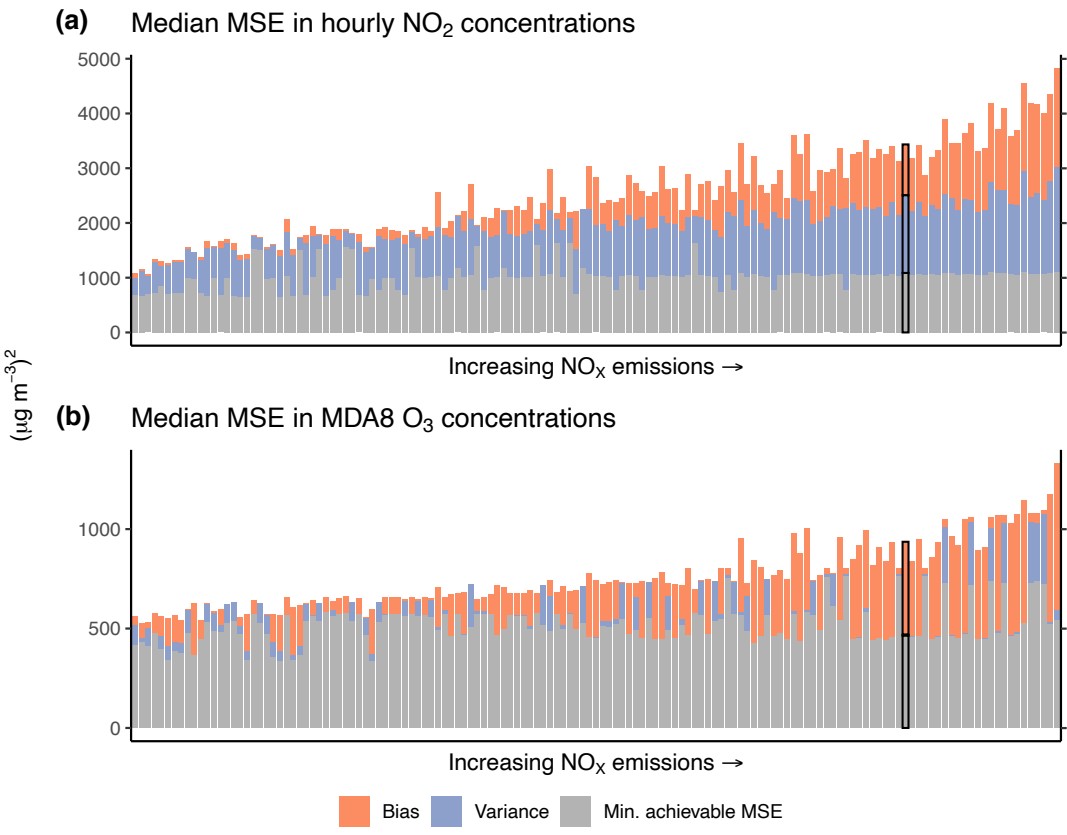

**Figure 6. Decomposed median mean square errors (MSE) in (a) hourly NO₂ concentrations and (b) daily maximum 8-hour mean (MDA8) O₃ concentrations associated with the optimised perturbed emissions ensemble (PEE) simulations, arranged in ascending order of the input annual total NOₓ emissions (from left to right), and the base run (marked by black frames).**


On account of the analysis above, we only used observations of $NO_2$ to constrain $NO_X$ emissions, which is also in line with numerous top-down emission optimisation studies using satellite observations of column $NO_2$. Figure 7(f) shows a strong positive correlation between the median MSE in hourly $NO_2$ and the value set for the parameter for transport sector $NO_X$ emissions in a simulation. Simulations with a median MSE within the 1st quartile are forced with transport emissions 6-65%

of those in the base emissions. This range continues to reduce (from both ends) with improving simulation performance, such that when the median MSE falls below the 10th percentile, the corresponding traffic emissions are only 7-43% of those in the base emissions. The range determined by the top performing 5% of the simulations (i.e. with a median MSE within the 5th percentile) remains the same, while it can be further constrained one-sidedly to 7-18% if only the top 1% were considered. However, as the top 1% of a 140-member ensemble contains (maximally) 2 simulations, the difference in whose average

performance is marginal (≤1.5%), this range was considered not robust.





In addition to the total magnitude, the night-time fraction of the transport sector $NO_X$ emissions could also be constrained (Fig. 7g). Instead of hourly $NO_2$, performance of the optimised PEE simulations was evaluated against the observed annual mean diurnal variations of $NO_2$ at each long-term monitoring site. The median of all 33 MSEs in the diurnal profiles at

individual sites modelled by a simulation was also used to represent its average performance in modelling the different diurnal profiles observed within the study area (see Fig. S2). Though not as evident as the case of total transport emissions, the range of the night-time fraction also becomes narrower with improving model skill. Amongst simulations with a median MSE in diurnal $NO_2$ profiles within the 15th percentile, 11-29% of the transport emissions occur at night (in contrast to the 9% in the base emissions). In the top 5% of the simulations, this fraction varies between 15% and 25%.


Emission parameters for other source sectors could not be constrained with strong confidence (Fig. 7a-e), as the ranges of parameter values only start to noticeably differ from the full uncertainty range when the median MSE in hourly $NO_2$ falls within the 10th percentile or even below. The residential sector is the smallest source sector of $NO_X$ in the base emissions (see Fig. 2a). A source apportionment of the base run reveals that its contribution to the annual mean $NO_X$ concentrations at

individual long-term monitoring sites varies between 3% and 8% (Fig. S5). It can thus be expected that even a 150% increase, i.e. its upper bound of uncertainty (see Table 1, column *Optimised PEE*), is not sufficient to cause substantial changes in the simulation performance for $NO_X$, based on which the emissions can be constrained.

Interestingly, Fig. S5 also reveals that at most sites, the contribution of $NO_X$ base emissions from the power sector to the

modelled annual mean $NO_2$ concentrations is even smaller than that from the residential sector, though the emissions are over two times higher (see Fig. 2a). This is attributable to the nature of the power sector, which is characterised by a few large point sources emitting at elevated levels. These sources undergo greater dispersion before reaching the surface-based monitoring sites (compared to ground level emissions), thus their impact at most sites is small. As an example, the annual mean $NO_X$ concentration simulated at the site NZG is very similar to that modelled at the site GC, yet less than 3% of this

concentration stems from $NO_X$ emissions from the power sector, as opposed to the 11% share at GC (Fig. S5). This can be explained by Fig. S6 in which the concentration resulted from power emissions is further apportioned to each grid cell. It is evident that the concentration at GC is predominantly contributed by the grid cell directly to its west. In fact, this grid cell contains the highest power emissions of $NO_X$ within the study area. In comparison, the site NZG is located at some distance from other grid cells with relatively large power sources. The contributions of these grid cells to the $NO_X$ concentrations at

NZG are smaller. The differences are about one order of magnitude, as the contributions are log transformed in the figure.

The fact that emissions released at higher levels are not well sampled by the existing surface-based monitoring sites also applies for those of $NO_X$ from industrial sources (emitted at up to 152 m). In addition, emissions from both source sectors were separated into two parameters and perturbed simultaneously (i.e. in an uncorrelated matter) for varying vertical





distributions. The uncertainty range of the sum of the two parameters, i.e. the total power or industrial emissions, was thus smaller than the individual uncertainty ranges and may not be sufficiently large to capture the actual emissions.

The short-term, independent SNAQ measurements of $NO_2$ are also used to constrain the emission parameters following the same approach (Fig. S7). Similarly, optimised PEE simulations showing better performance for hourly $NO_2$ concentrations and mean diurnal variations of $NO_2$ concentrations (over the measurement period) at SNAQ sites are associated with lower transport sector total emissions, but a higher percentage of these emissions occurring at night. In the top performing 5% of the simulations, total traffic emissions are 7-43% of those in the base emissions, while the night-time fraction varies between 15% and 26%. The fact that the uncertainty ranges constrained by the short-term SNAQ measurements are consistent with those using long-term reference measurement demonstrates the robustness of both this approach and the findings. This also supports the use of low-cost sensors for this particular application, as they are more affordable for deployment in a dense network.







**Figure 7.** Average performance of the optimised perturbed emissions ensemble (PEE) simulations and the base run (marked with black strokes) as a function of emission parameter values. The scales on the x-axes correspond to the uncertainty ranges in Table 1, column *Optimised PEE*. The top performing 25%, 20%, 15%, 10%, 5% and 1% of the simulations are coloured in a darkening green shade, as measured by their median mean square errors (MSE) in hourly NO₂ concentrations at the long-term monitoring





**sites across the modelling domain (see Fig. 6) in all panels except in (g), where median mean square errors in the annual mean diurnal variations of NO$_2$ concentrations are used (note the different scale on the y-axis).**


## 4 Discussion

According to the base emissions, total NO$_X$ emissions from the transport sector were $2.1\times10^5$ Mg within the study area which extends over most of Beijing and parts of Hebei Province in 2013, 9% of which occurred during 0-5 am. Based on the top 5% of the optimised PEE simulations for modelling NO$_2$ concentrations and diurnal profiles, we found that transport

NO$_X$ emissions were likely to have decreased to $1.5\text{-}9\times10^4$ Mg (i.e. a 57-93% reduction) in 2016 and the night-time fraction was between 15% and 25%.

An exact comparison of these results with findings of previous studies is not possible, as the emissions were investigated over different spatial and/or temporal scales. However, it is possible to compare the relative changes in emissions. Biggart et

al. (2020) found that total NO$_X$ emissions rates from the same *a priori* emissions inventory (and all source sectors) were 1.8 times higher than those from an optimised emissions inventory (with which NO$_X$ and NO$_2$ concentrations in much better agreement with the corresponding observations were simulated by ADMS-Urban), though their investigation was focused on a small domain in urban Beijing and the duration of the APHH-Beijing winter campaign only. They also found that the modelled mean diurnal variations in NO$_2$ concentrations at most sites could be substantially improved when the night-time

fraction of NO$_X$ emissions was increased by 25% and 50%. Nonetheless, as mentioned in Sect. 2.2, this was defined as NO$_X$ emitted between 11 pm and 6 am from all source sectors.

Squires et al. (2020) compared NO$_X$ emission rates in the same *a priori* emissions with flux measurements made from a tower in central Beijing (where SNAQ39 was also deployed, see Fig. 1) during the APHH-Beijing winter campaign. The

study area was also smaller than that in this work, as the flux footprint (i.e. the upwind source area of the measured fluxes) was on average within 2 km (maximal 7 km) of the tower. Compared to the measured fluxes, the emissions rates were found to be overestimated by a mean factor of 9.9. They further considered emissions only from the transport and residential sectors (as no industrial or power sources were identified within the average footprint) and reduced these by 30%, yet these were still on average 3.3 times higher than the fluxes. They also found much smoother diurnal variations in the NO$_X$ fluxes

compared to those in the estimated emission rates, indicating that the night-time fraction was underestimated in the latter.

In the standard MEIC v1.3 (from which the *a priori* emissions were downscaled and re-gridded), annual NO$_X$ emissions from the transport sector were estimated to be $1.05\times10^5$ Mg in Beijing and $6.59\times10^5$ Mg in Hebei Province in 2013. In 2016, these figures decreased to $8.87\times10^4$ Mg and $5.68\times10^5$ Mg, respectively, corresponding to reductions by 15.5% and 13.8%,





which are substantially lower than the reductions reported in this work. A slightly larger reduction of 20% (from $1.44 \times 10^5$ Mg in 2013 to $1.15 \times 10^5$ Mg in 2016) was estimated for vehicle (including on- and off-road vehicles) sources of $NO_X$ in Beijing by Cheng et al. (2019), who used a bottom-up emissions inventory for Beijing which was compiled from the county level and associated with finer spatial resolutions than MEIC established from the province level. They also concluded that vehicle emission control measures contributed the most to the total reductions in $NO_X$ emissions over this period. According

to the *China Vehicle Environmental Management Annual Reports*, $NO_X$ emissions from on-road vehicles were $7 \times 10^4$ Mg in Beijing and $5.2 \times 10^5$ Mg in Hebei Province in 2013. These fell by 14% and 10%, respectively, to $6 \times 10^4$ Mg and $4.7 \times 10^5$ Mg in 2016, despite an increase in vehicle ownership (Ministry of Environmental Protection of the People's Republic of China, 2014, 2017).

The downward trends in traffic sources of $NO_X$ between 2013 and 2016 found by the studies and reports mentioned above and this study contrast with the results in Xue et al. (2020). Based on yet another bottom-up emissions inventory for Beijing, they showed that mobile (i.e. on-road and off-road vehicles) sources of $NO_X$ increased from $9.6 \times 10^4$ Mg to $1.1 \times 10^5$ Mg over the same period. Though they found a 20% reduction in the total anthropogenic $NO_X$ emissions from 2013 to 2016, this was primarily attributed to optimised energy structure in the industry, power and residential sectors. Amongst the top performing

5% of the optimised PEE simulations, the reduction in total $NO_X$ emissions from the base emissions varies between 59% and 74%. These conflicting findings again highlight the presence of inherent uncertainties in emissions inventories, the impact of which on the results of this work is discussed in the following, along with the impact of other sources of uncertainty.

Uncertainties associated with the base emissions are twofold. Inherent uncertainties due to underlying emission factors and

activity rates were estimated to be $\pm 31\%$ for $NO_X$ emissions in MEIC (Cheng et al., 2019), which are smaller than the uncertainty ranges in Table 1, suggesting that the trends in (real-world) emissions from 2013 to 2016 represent a larger source of uncertainty when using the 2013 base emissions for 2016 simulations. Additional uncertainties may have been introduced when the standard MEIC v1.3 was downscaled and re-gridded to the *a priori* emissions inventory used in this work. As an example, Zheng et al. (2017) compared MEIC with another emissions inventory with a much larger share of

point sources (which were allocated directly to grid cells) and different sets of spatial proxies to allocate non-point sources. They found that $NO_X$ emission fluxes from the most populated grid cells in Hebei Province were overestimated by 46-140% in MEIC, mainly driven by spatial proxies that over-allocated industrial emissions to urban areas.

While these two types of uncertainties may be non-negligible in the base emissions, we believe that they are not fully

propagated into uncertainties in the updated emission estimates. This is because the PEE approach does not require the *a priori* emissions to be bias-free. Though most emission parameters were defined relative to the corresponding values in the base emissions for an efficient perturbation, their uncertainty ranges were ultimately constrained solely by the observations. Meanwhile, uncertainties in the observational constraints are mostly small, which is demonstrated by the consistency in the



results derived using two independent sets of observations. Nonetheless, it is worth noting that spatial inhomogeneities of the
biases in the base emissions are propagated into the updated emission estimates, as spatially uniform perturbations were applied when constructing the optimised PEE.

Another source of uncertainty is the input lateral boundary conditions which include meteorology and background pollution levels. The impacts of uncertainties in the input meteorological observations (i.e. due to measurement errors) are minimal, as
reported in Yuan et al. (2021) using simulations forced with perturbed meteorological data. The effect of uncertainties associated with different background concentrations of $NO_X$ and $O_3$ on the constrained $NO_X$ emissions is discussed next in the context of uncertainties in the underlying chemical mechanisms in ADMS- Urban.

The chemical partition of $NO_X$ emissions into $NO_2$ concentrations by the model represents a potentially important source of
uncertainty. Many studies that infer $NO_X$ emissions from satellite observations of the tropospheric $NO_2$ column assumed an accurate representation of $NO_X$ chemistry in the CTM used. However, Valin et al. (2011) demonstrated the presence of biases in the modelled $NO_2$ column that are dependent on the horizontal resolution of the model, which has implications on the inference of $NO_X$ emissions when matching the modelled column to satellite observations. These biases result from an inaccurate representation of $NO_2$ lifetime (and thus concentration) at coarse resolutions, which is determined primarily by
OH concentration in daytime which, in turn, has a non-linear dependence on $NO_X$ concentration.

When $NO_2$ rapidly interconverts with NO via reactions with $O_3$ and peroxy radicals in the presence of sunlight, it is also oxidised by hydroxyl radical (OH) to nitric acid ($HNO_3$) which can then be removed from the atmosphere via wet/dry deposition. At night, NO cannot be replenished as no $NO_2$ photolysis takes place. With very low concentrations of OH, $NO_2$
is mainly oxidised by $O_3$ to form nitrate radical ($NO_3$) which further reacts with $NO_2$ and reaches an equilibrium with dinitrogen pentoxide ($N_2O_5$). $NO_2$ also hydrolyses on aerosol surfaces to form nitrous acid (HONO). $NO_3$, $N_2O_5$ and HONO are described as night-time reservoirs of $NO_X$, as they can regenerate NO or $NO_2$ after sunrise. These reservoir species are absent in the semi-empirical chemical mechanism described in Sect. 2.3.

The SNAQ measurements which included NO allowed an evaluation of the $NO_X$ photolytic chemistry in ADMS-Urban. Han et al. (2011) demonstrated the presence of strong linear relationship between $NO_X$ and NO during night-time and between $NO_X$ and $NO_2$ during the day, in observations from Tianjin, another megacity not far from Beijing. Following this approach, simple linear regression models were fitted to the SNAQ measured (hourly averaged) mixing ratios of NO and $NO_2$, as a function of the corresponding mixing ratios of $NO_X$. In addition, a linear function was fitted between log transformed mixing
ratios of $O_3$ and (non-logarithmic) mixing ratios of $NO_X$ due to a negative correlation between the two (Fig. S8a). Additional linear functions were also fitted to daytime or night-time data only. Data from the individual SNAQ sites were not differentiated because of the short duration of operation at each site. Consistent with findings by Han et al. (2011), a strong




correlation can be found between the daytime mixing ratios of $NO_2$ and $NO_X$. The correlation between $O_3$ and $NO_X$ is also stronger during the day (but weaker than that between daytime $NO_2$ and $NO_X$). The highest coefficients of determination are

associated with the three models (i.e. using day-time, night-time or all data, respectively) for NO as a function of $NO_X$, the differences between which were minimal.

We then fitted linear models to the corresponding output (i.e. for the same locations and time frame) from the top performing 5% of the optimised PEE simulations (see Sect. 3). These are compared with models fitted to the SNAQ measurements with

respect to the slope, as it indicates the amount of changes in NO, $NO_2$, (log transformed) $O_3$ as $NO_X$ increases/decreases, which is directly related to the input $NO_X$ emissions. As can been seen from Fig. 8a, the modelled slopes of the linear models between $NO_2$ and $NO_X$ are higher, whereas modelled slopes for NO as a function of $NO_X$ are lower compared to those observed, irrespective of the time of day. The discrepancies between modelled and observed slopes for $O_3$ as a function of $NO_X$ are relatively small. This suggests that with similar concentrations of $NO_X$ as observed, the top performing optimised

PEE simulations tend to overestimate $NO_2$ while underestimating NO. In other words, lower $NO_X$ emissions may be needed for the model to simulate $NO_2$ concentrations that are consistent with the observations. This suggests that the constrained emission estimates of $NO_X$ are indeed sensitive to the $NO_X$ photolytic chemistry in ADMS-Urban and in this case, may be low-biased.

There are several possible explanations for the model's tendency to partition more (less) $NO_X$ into $NO_2$ (NO). As described in Sect. 2.3, reactions with $O_3$ (R4) and organic radicals (R2) are the two major pathways of NO oxidation to form $NO_2$ in ADMS-Urban. Organic radicals are produced via reaction (R1) in which concentrations of the composite ROC are defined as TVOC concentrations multiplied by a reactivity coefficient. In ADMS-Urban version 4.2 (used in this study), a coefficient of 0.1 is adopted. It is set to 0.05 in the most up- to-date version 5 (Cambridge Environmental Research Consults Limited, 2017,

2020). As a secondary pollutant, modelled $O_3$ concentrations depend on the input background levels, for which there is no widely accepted definition. To investigate the sensitivity of the model output to these two NO oxidation pathways, a series of sensitivity simulations were performed. All simulations were input with the same $NO_X$ emissions as R97, the optimised PEE simulation with the best performance in simulating hourly $NO_2$ concentrations at the long-term monitoring sites. Again, simple linear regression models were fitted between hourly mixing ratios of NO, $NO_2$, (log transformed) $O_3$ and $NO_X$ output

by these simulations.

Figure 8b shows that different definitions of background $NO_2$ and/or $O_3$ (see Table S3) indeed have an impact on the modelled NO-to-$NO_2$ conversion. For example, in the sensitivity simulation S5 input with the lowest background levels of $NO_2$ and $O_3$, the modelled slopes for $NO_2$ as a function of $NO_X$ are considerably lower than the corresponding slopes

modelled by R97, and thus closer to the observed slopes. Also, the slopes between NO and $NO_X$ associated with outputs from S5 are higher than those found in outputs from R97 and agree better with the observed slopes. However, it is important



to underscore that this does not suggest that the $10^{th}$ percentile concentration is most representative of the background concentrations of $NO_2$ and $O_3$. It simply highlights the impact of the input background concentrations of reactive pollutants on the model outputs of relevant species, and thus on the emission estimates inferred on the basis of these model outputs. Also, it is worth noting that this chemistry is also influenced by the input background levels of NO. However, the specific sensitivities were not investigated, as the upwind concentrations extracted from the CAMS reanalysis dataset are the only set of NO observations available long-term.

Similarly, it is not unexpected that using a reactivity coefficient of 0.05, effectively halving the ROC available in R97 (which results from both the input VOC emissions and their background concentrations), less $NO_X$ is partitioned to $NO_2$. In contrast, setting the reactivity coefficient to 0.2 leads to an even more pronounced overestimation of $NO_2$, accompanied by an underestimation of NO. The modification of this reactivity coefficient in ADMS-Urban version 5 is thus supported. Unlike background pollutant concentrations, however, the effect of ROC concentrations on model chemistry is restricted to daylight hours, as they only produce radicals in the presence of solar radiation.

Figure 8. Slopes of linear regression models fitted between the modelled hourly mixing ratios of NO, NO$_2$, log transformed O$_3$ and those of NO$_X$ at all SNAQ sites by (a) the top performing 5% of the optimised perturbed emissions ensemble simulations (PEE) and the base run, (b) the background concentration sensitivity simulations and the best PEE simulation R97, (c) the ROC concentration sensitivity simulations and the best PEE simulation R97, compared to the corresponding slopes fitted to SNAQ measurements. Details of the background concentration sensitivity simulations are provided in Table S3. The simulations Si and Sii represent a doubling and halving of the ROC concentrations (by modifying the reactivity coefficient) in R97, respectively. In all panels, day-time is defined as complete hours between sunrise and sunset in Beijing during November-December 2016, namely 8:00-15:00 local time.



Another factor that affects the modelled NO$_X$ photolytic chemistry is the f-NO$_2$ in the input emissions (see Sect. 3.1). Although not investigated with sensitivity experiments, it can be expected that a higher f-NO$_2$ would result in higher NO$_2$ concentrations and lower NO concentrations simulated by the model, thus further increasing the discrepancies between the modelled and observed slopes of linear functions, whilst a lower f-NO$_2$ would have the opposite effect.

## 5 Conclusions

We developed a novel approach to update *a priori* emission estimates using ground-based network measurements as constraints and an ensemble of forward simulations which are input with a PEE. Using this approach, we were able to update the transport sector NO$_X$ emissions in Beijing from a 2013 emissions inventory for the year 2016. The updated emissions are substantially lower with a higher proportion occurring at night-time and are broadly consistent with findings of several previous studies. It would be possible to also update emissions from other (non-negligible) source sectors, provided that appropriate measurements were available.

As with existing emission optimisation techniques, this approach is sensitive to the chemical mechanisms in the underlying model, the uncertainties of which can be propagated into uncertainties in the emission estimates. Nonetheless, this approach has several unique advantages. Compared to inverse modelling techniques, the construction of a PEE and the forward simulations are rapidly executable. Even when the Gaussian dispersion model used in this study is replaced with a CTM for more explicit representations of chemistry, the efficiency can be maintained via parallel computing. Also, surface-based measurements of ambient concentrations are used as constraints, which are readily available and closer to the sources of emissions than the satellite-based measurements. This proximity to emission sources may be particularly important for capturing the high temporal and spatial variability of highly reactive species. For example, Qu et al. (2021) found that in comparison to surface concentrations, the NO$_2$ column showed a muted response to the step decrease in NO$_X$ emissions in the US during the COVID-19 crisis. Most importantly, this approach allows for an update of emissions by source sector, which is more relevant for policy interventions than total emissions, as they directly reflect the (in)effectiveness of the corresponding pollution control measures. Hence, we believe that this approach, particularly combined with low-cost sensors, has great potential in providing timely updates of emissions in regions undergoing rapid changes, where emissions inventories may be biased or out-dated as soon as they have been compiled and computing facilities may be limited.



## Code and data availability

Codes used to generate the perturbed emissions ensemble using the R language are available at: https://github.com/yuanle731/PEE. The Multi-resolution Emission Inventory for China (MEIC) is available upon request at http://www.meicmodel.org. Long-term air quality monitoring data from Beijing is archived at https://quotsoft.net/air/ and http://data.epmap.org.

## Author contribution

LY developed the methodology with advice from AA and performed and analysed the model simulations with advice from AA, OP, RJ, CH, and DC. OP and RJ provided the low-cost sensor measurements and advised on data processing. CH and DC provided the licence of ADMS-Urban and advised on the simulation setup. QZ provided the Multi-resolution Emission Inventory for China. LY prepared the manuscript with contribution from all co-authors.

## Competing interests

Some authors are members of the editorial board of journal Atmospheric Chemistry and Physics. The authors have no other competing interests to declare.

## Acknowledgements

We thank Oliver Wild, Michael Hollaway and Michael Biggart for processing the base emissions. Thanks also go to Sue Grimmond and Simone Kotthaus for providing mixed layer height measurements and comments on the manuscript.

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
