# Peer review of "Improving NOx emissions in Beijing using network observations and a novel perturbed emissions ensemble"

_Atmospheric Chemistry and Physics, 2022_

## Author Comment (AC1)

We are very grateful to both reviewers for their constructive feedback. We have revised the manuscript and hope that the revisions address the reviewers' comments. In the following, each comment and our corresponding response are sequentially listed, in **bold** and plain text, respectively. Text in the original manuscript is shown in red with strikethrough marking deletions. New text in the revised manuscript is coloured blue. We will be submitting two versions of the revised manuscript: a clean version and a version with tracked changes.

| ANONYMOUS REFEREE #1 | 1 |
|----------------------|---|
|                      | Т |

- ANONYMOUS REFEREE #2......15

**Anonymous Referee #1**

- Title: I am not sure what the novelty in the perturbed emission ensemble is. Thus, I suggest to remove "novel" from the title.

We thank the reviewer for this comment. Also taking the suggestion from Referee #2, the title now reads:

Improving NOx emissions estimates in Beijing using network observations and a novel perturbed emissions ensemble

- The evaluation of the ensemble members is mainly based on the MSE. However, I was wondering if the ensemble shows a sign-change in the bias of NO2 concentrations, which would further support the estimation of the optimal emission data.

We agree with the reviewer that the MSE does not indicate the sign of model biases. Thus, we used the normalised mean bias factor (NMBF) in the preliminary evaluation of both the initial and the adjusted PEE simulations. The results are now presented in Fig. S2 in the revised Supplementary Information (whereby panel (a) corresponds to Fig. S1 in the original Supplementary Information) and are also shown below. Panel (a) reveals that at 22 long-term monitoring sites, the initial PEE simulations fail to output NO2 concentrations comparable to the observations. Specifically, NO2 concentrations at 19 sites are overestimated by the entire ensemble, while those at 3 other sites are underestimated. This widespread positive bias prompted us to decrease the lower bounds of uncertainty (and thus widen the uncertainty ranges) for most emission parameters. As shown in panel (b), the adjusted PEE simulations are associated with larger ranges of NMBFs. At 3 more sites, the range of NMBFs crosses the dashed line representing zero bias, indicating that more NO2 concentrations comparable to the observations are simulated with the adjusted PEE.

NMBF in annual mean NO2 concentrations

Figure S1S2. Distribution of the normalised mean bias factors (NMBF) in annual mean NO2 concentrations associated with (a) the initial and (b) the adjusted perturbed emissions ensemble (PEE) simulations and the simulation forced with the base emissions at each long-term monitoring site. In each panel, the simulation forced with the base emissions is also shown. Note that as different background levels of NO2 and O3 are input (in accordance with the initial and the adjusted PEE simulations), these two simulations are not identical, also indicated by the different NMBFs. The monitoring sites are colour-coded according to the site type: urban site (magenta), traffic monitoring site (purple), suburban site (orange), clean site (light green) and regional background site (green). The circle at the leftmost end of the boxplot for YLD in panel (b) represents an outlying PEE simulation (i.e. with a NMBF outside 1.5 times the interquartile range below the lower quartile).

- I understand that the choice of spatiotemporal uniform emission perturbations suggests an evaluation of averaged concentrations over all stations. An evaluation at single stations was initiated by e. g. Fig. 5, but I would have expected a more detailed investigation of the ensemble members in different regions. Potentially, the results allow also for a spatially heterogeneous emission correction.

We thank the reviewer for this comment. As our conclusions were drawn with respect to annual emissions within the modelling domain, we think an evaluation of the PEE simulations' average performance across all measurement sites is appropriate. We agree with the reviewer in that Fig. 4 and Fig. 5 reveal disparities in the simulations' performance at individual sites, suggesting the presence of spatial biases in the *a posteriori* emission estimates derived from these simulations. If gridded *a posteriori* emission estimates were to be derived, spatially heterogeneous perturbations to the prior emissions would be ideal, and the evaluation of the simulations would be best performed locally at individual sites within certain localisation scale. We think these are beyond the scope of this study but are important directions for future development of our method.

- Although VOC emissions (and background concentrations) are included in the model, the impact on these emissions and potential uncertainties is not addressed adequately. Especially in high NOx concentrations, the O3 concentrations depend highly on the available VOC. The manuscript only considers NOx emissions as main source of uncertainty. A discussion on the impact of this choice is appreciated.

We agree with the reviewer that emissions and background concentrations of VOCs have an impact on the NOX-O3 chemistry and thus the *a posteriori* emission estimates of NOX. Ideally, VOC emissions should be perturbed and constrained along with NOX emissions. This was not possible for two reasons. On one hand, VOC concentrations participate in the NOX photolytic chemistry scheme in ADMS-Urban as a source of radicals, but are not themselves affected by the chemical calculations. On the other hand, long-term network observations of VOC concentrations were unavailable. Hence, we discussed the impact of VOCs on the derived NOX emissions qualitatively using the two ROC concentration sensitivity simulations described in Section 4. This may not have been clear in the original manuscript. The relevant paragraph is revised as follows:

The effect of organic radicals on the partition of  $NO_X$  between  $NO_2$  and NO is shown in Fig. 8c by varying the concentrations of ROC. As explained above, ROC concentrations are controlled by both the TVOC concentrations (that result from the input emissions and background levels of VOC) and a reactivity coefficient which was set to 0.1 in R97 (as with other adjusted PEE simulations). Similarly, it is not unexpected that using a reactivity coefficient of 0.05, effectively halving the ROC available in R97 (which results from both the input VOC emissions and their background concentrations), less NOx-is partitioned to NO2. In contrast, setting the reactivity coefficient to With fixed TVOC concentrations, using a coefficient of 0.2 doubles the ROC available to produce HO2 and RO2, leading<del>leads</del> to an even more pronounced overestimation of NO2, accompanied by an underestimation of NO. In contrast, halving the ROC concentrations by using a coefficient of 0.05 partitions less of the NOx emitted into NO2. This highlights that the emissions and background concentrations of VOC (which are not evaluated in this study due to a lack of observations) also have an impact on the modelled NOx photolytic chemistry and thus the *a posteriori* emission estimates of NOx. It is also worth noting that biogenic VOCs are likely underestimated in the current simulations, as these are only represented by one of the 8 species (i.e. isoprene) output by the CAMS reanalysis product used to approximate the background levels of TVOC, and are not represented at all in the base emissions (which include anthropogenic sources only). Despite low in concentrations in the study area (compared to anthropogenic VOCs) (Mo et al., 2018), they are associated with high radical production and thus O3 creation potentials. The modification of this reactivity

coefficient in ADMS-Urban version 5 is thus supported. Unlike background pollutant concentrations of NOX and O3, however, the effect of ROC concentrationsVOCs on the modelled NOX-O chemistry is restricted to daylight hours, as they only produce radicals in the presence of solar radiation in the model.

- A discussion on the representativity of observation sites (especially urban and traffic) is required. Is the model resolution sufficient to be compared the traffic measurement stations? We thank the reviewer for this comment. We agree with the reviewer that the gridded emissions dilute traffic emissions within each grid cell, which could result in an underestimation of roadside NO2 concentrations. In fact, this was found at some sites in Beijing in ADMS-Urban simulations input with another emissions inventory of the same 3 km  $\times$  3 km resolution (Biggart et al., 2020). However, as our conclusions were drawn with respect to annual traffic emissions within the domain using measurements of all 33 sites as constraints, we believe this is a minor issue, since potential underestimation at traffic monitoring sites would be accompanied by overestimation at other sites at some distance. In comparison, we think the under-representativeness of the existing site network for the power and industrial emissions is a major limitation. We have included a short discussion on this in Section 4:

Meanwhile, uncertainties Uncertainties in the observational constraints are twofold. While those due to measurement errors are mostly likely small, which is as demonstrated by the consistency in the results derived using two independent sets of observations, the underrepresentation of the existing observations of power and industrial sources prohibited an update of emission strengths from these sources.

- the wording initial PEE and optimized PEE is somehow misleading. Only as I have finished section 2 I have understood that this approach is not a data assimilation or inversion method. Maybe "adjusted PEE" instead of "optimized PEE" would be clearer. In the context of observations, "optimized" always feels like there is some optimization method applied, which is certainly not the case in this manuscript.

We thank the reviewer for this suggestion. The term "adjusted PEE" is now used instead of "optimised PEE" throughout the revised manuscript.

- To the simulation setup: It is not really clear, which simulations have been done. There is a base run with additional 140 member ensemble with perturbed NOx emissions. However, the simulation episode should be state here explicitly (am I right that the full year 2016 was simulated?), also the model resolution (horizontal and vertical) is missing. A link to the discussion section, where the limitations introduced by model simplifications is discussed, would be good. It would have been easier for the understanding of the results that only the optimized PEE is used for the simulations.

We thank the reviewer for these comments. We did not mention exact spatial resolutions of the simulations, as we did not output pollutant concentrations in a gridded format, but only those at measurement locations. More explanations on the possible output resolution of the

model are provided below in our response to the reviewer's comment on line 233 (in the original manuscript). The total number and the time frame of the simulations performed are added in the last paragraph under Section 2.3, where it is also made clear that only the adjusted PEE simulations (and not the initial PEE simulations) were used to derive *a posteriori* emission estimates:

The input meteorology data and background pollutant concentrations described above provided the same lateral boundary conditions for all optimised the 140 adjusted PEE simulations, among which only the emissions of NOx (and NO2) varied. An additional simulation forced with these boundary conditions and the base emissions was also performed and is hereinafter referred to as the base run. All 141 simulations were run for the whole year of 2016 to produce hourly pollutant concentrations at each measurement location (see Fig. 1). Output of these simulations were then compared to measurements to derive *a posteriori* emission estimates.

**- line 52: A discussion on the local anthropogenic and biogenic share of NOx (and esp. VOC) emissions would be appreciated. Are biogenic emissions in this regions (especially in summer) negligible compared to the anthropogenic emissions?**

We thank the reviewer for this valuable input. We have revised this paragraph to include a short discussion on the relative importance of anthropogenic and soil  $NO_x$  emissions in China, as the latter represents a larger natural source compared to lightning (Lin, 2012).

Though-NOx can be produced from both anthropogenic and natural/biogenic sources such as fossil fuel combustion, biomass burning, soil microbial processes and lightning<del>, they are predominantly released from anthropogenic sources including fossil fuel combustion and open biomass burning (Feng et al., 2020;</del> Lee et al., 1997). Global total anthropogenic NOx emissions flattened around 2008, as reductions in Europe and North America were offset by increases in Asia (Hoesly et al., 2018). China, in particular, witnessed a rapid rise in anthropogenic NOx emissions until 2011-2012...stricter vehicle emission standards combined with accelerated fleet turnover (Liu et al., 2020). Decreases in anthropogenic sources are accompanied by an increased importance of soil NOx emissions, which are largely driven by nitrogen fertiliser application and can reach up to 20% of the anthropogenic emissions in the crop growing season in some regions with high agricultural activities (Lu et al., 2021). These emissions are relatively poorly quantified and currently unabated (State Council of the People's Republic of China, 2018).

However, for our study area centred around Beijing, and on an annual scale (as are the derived emission estimates), we believe that soil  $NO_X$  emissions are negligible compared to anthropogenic  $NO_X$  emissions. This is also supported by the Supplementary Fig.S1 in Lu et al. (2021) which shows Beijing dominated by "high anthropogenic  $NO_X$  emission" model grids. Hence, soil  $NO_X$  emissions are not discussed further in the revised manuscript.

For a discussion on the impact of VOCs, the reviewer is kindly referred to our response to the reviewer's 4th comment, where we have also included a discussion on the likely underrepresentation of biogenic VOCs in the simulations.

**- Line 56-58: Citations for the different action plans required**

References to the action plans are added as:

Emission reduction targets were first announced in the 12th Five-Year Plan (2011-2015) (People's Republic of China, 2011), followed by the Action Plan on Prevention and Control of Air Pollution (2013-2017) (State Council of the People's Republic of China, 2013) and the Three-Year Action Plan for Winning the Blue Sky Defence Battle (2018-2020) (State Council of the People's Republic of China, 2018).

**-line 64: ...method, which... (add a comma)**

The sentence is modified to read:

Some have used a bottom-up method which that combines specific emission factors (i.e. mass of a pollutant emitted per unit fuel consumption or industrial production) energy consumption data from individual source sectors with the corresponding emission factors activity rates (i.e. fuel consumption or industrial production),...

**-line 65/66: I guess you are talking about the amount of studies investigating emission data, please be more precise: which data? What is the large amount of the data? How can data solve the time-lag issue?**

Here we were referring to the large amount of input data (e.g. emission factors and energy consumption data) required to establish a bottom-up emissions inventory, most of which are not available real-time. Thus, the collection and compilation of these data result in a considerable time-lag in the occurrence of emissions inventories, typically one to several years. The text is modified as follows to avoid ambiguity:

However, The substantial amount of the underlying data are mostly not immediately available, required not only results resulting in an inevitable time-lag between the occurrence of emissions and the establishment of an inventory (Janssens-Maenhout et al., 2015).

**- line 68: It may be worth elaborate on emission uncertainties and their impact**

We thank the reviewer for this comment. A reference for the uncertainty estimates for MEIC is provided in the 6th paragraph under the discussion section. Two references for the impact of emission uncertainties are added here:

...large and poorly quantified uncertainties into the emission estimates (Hong et al., 2017; Zhao et al., 2011)-, which can be further propagated through modelled pollutant concentrations into disease or mortality burden (Crippa et al., 2019) and economic loss estimates (Solazzo et al., 2018).

- It would be worth elaborate more about the pros and cons of the different methods you are summarizing in the introduction. Why are you proposing the new method, what is the strength of your method compared to the other methods?

We thank the reviewer for this comment. We have amended this paragraph such that our discussion of the pros and cons of the bottom-up and top-down methods stands out more clearly. The strength of our method is summarised in a sentence at the start of the next paragraph:

Some have used a bottom-up method which that combines specific emission factors (i.e. mass of a pollutant emitted per unit fuel consumption or industrial production) energy consumption data from individual source sectors with the corresponding emission factors activity rates (i.e. fuel consumption or industrial production), thus providing sector- or process-resolved emission estimates (Liu et al., 2016; Zhang et al., 2009; Zhao et al., 2013; Zheng et al., 2018). However, The substantial amount of the underlying data are mostly not immediately available, required not only results resulting in an inevitable time-lag between the occurrence of emissions and the establishment of an inventory (Janssens-Maenhout et al., 2015), Moreover, it also propagates they can introduce potentially large and poorly quantified uncertainties into the emission estimates (Hong et al., 2017; Zhao et al., 2011), which can be further propagated through modelled pollutant concentrations into disease or mortality burden (Crippa et al., 2019) and economic loss estimates (Solazzo et al., 2018). Other studies have inferred topdown estimates of emissions using satellite observations due to their continuous spatiotemporal coverage and near real time availability...four-dimensional variational assimilation (Kurokawa et al., 2009) have been increasingly adopted to combine satellite observations and CTM simulations with prior emission estimates to derive a posteriori emission estimates. These inverse methods-are time consuming and computationally demanding provide more timely emission estimates of high spatial and temporal coverage (based on the nature of satellite observations). Nonetheless, the derived emission estimates are not resolved by source sector. The a posteriori emission estimates They are also subject to uncertainties propagated from the satellite retrievals or and the model simulations...

This study introduces a novel approach that provides timely updates of a priori emission estimates by source sector using readily available in-situ air quality observations. Using this approach, is aimed at optimising a priori NOx emissions in a bottom-up inventory compiled for Beijing for the year 2013 to account for emissions in are updated for 2016 using a novel approach.

**-line 83 – 86: This statement is not only valid for satellite data. Insufficient chemistry always influences the model results and, thus, the analysis.**

We agree with the reviewer that any analysis involving model simulations can be affected by insufficient chemistry pathways in the model. Here we were referring specifically to the effect of model chemistry on the inversely derived emission estimates. The text is revised to avoid ambiguity:

For instance, Archer-Nicholls et al. (2021) showed large differences in the NO2 column density simulated withby two chemical mechanisms with different treatment of non-methane volatile organic compounds (NMVOCs), which are integrated into the same model with identical NOx emissions, which resulted from different treatment of non-methane volatile organic compounds (NMVOCs) and thus the conversion of NOx to sinks and reservoir species (via

reactions with oxidation products of NMVOCs). When used in inverse modelling, these modelled  $NO_2$  quantities would result in different *a posteriori*  $NO_x$  emissions.

**-line 106 -111: I feel like this is too much detail for the manuscript. Is it necessary to follow the study to know the accuracy of the measurement instruments?**

- line 116-118: Also, is this information necessary for the manuscript? I don't feel so. Are the low-cost sensors influenced by a systematic error (bias) that may have an influence on the comparison?

We thank the reviewer for these two comments. The low-cost sensors were co-located with reference instruments prior to deployment and a systematic bias was not found. The fact that the *a posteriori* emission estimates derived independently using reference instrument measurements and low-cost sensor measurements are consistent also indicates minimal influence of measurement errors. These lines are now deleted.

- Table 1: Please include the night time definition for the initial and optimized ensemble in the caption. Also, the caption states night time fractions are in %, but values show ratios, please revise.

We thank the reviewer for this comment and pointing out the error. The definitions for the night-time fraction of transport sector NOx emissions in the two PEEs are included in a footnote, and the respective minimum and maximum values adopted are corrected to read: Table 1. Emission parametersa and the respective uncertainty ranges sampled by the initial and the optimised adjusted perturbed emissions ensembles (PEEs).

| Parameter                                               | Initial
PFF |                | Optimised
Adjusted |                |
|---------------------------------------------------------|----------------|----------------|-----------------------|----------------|
|                                                         |                |                | PEE                   |                |
|                                                         | Min            | Max            | Min                   | Max            |
| Industry sector ground level NO x emissions  | 0.4            | 1.6            | 0.05                  | 1.6            |
| Industry sector elevated NO x emissions      | 0.4            | 1.4            | 0.05                  | 1.4            |
| Power sector NO X emissions b     | 0.2            | 1.4            | NA                    | NA             |
| Power sector NO x emissions below 152 m      | NA             | NA             | 0.05                  | 1.6            |
| Power sector NO x emissions above 152 m      | NA             | NA             | 0.05                  | 1.6            |
| Residential sector NO x emissions            | 0.4            | 1.5            | 0.05                  | 1.5            |
| Transport sector NO x emissions              | 0.4            | 2              | 0.05                  | 1.5            |
| Night-time fraction of transport sector NO X | <del>0.1</del> | <del>0.4</del> | <del>0.1</del>        | <del>0.3</del> |
| emissions c                                  | 10             | 40             | 10                    | 30             |

a All parameters are defined as ratios of the 2016 emissions to the base emissions from 2013, except for the night-time fraction of transport sector  $NO_X$  emissions which is defined as a percentage (%) of the daily totals in 2016.

b Power sector NOX emissions are effectively represented by one parameter in the initial PEE. In the <del>optimised</del> adjusted PEE, the emissions are split into two parameters, namely emissions below and above 152 m.

c Night-time fraction of transport sector NOX emissions is defined as those occurring during

11am-6am in the initial PEE. In the adjusted PEE, it is modified to  $NO_X$  emitted during 0-5 am from the transport sector.

**-Line 178-179: Please include the number of experts that contributed to the poll.**

We thank the reviewer for this comment. More information about the expert elicitation is included:

An online questionnaire was designed for the elicitation (available at https://cambridge.eu.qualtrics.com/jfe/form/SV\_3eGxf9XvC7WXESV, last accessed 14 April 2022) and circulated via the mailing list of the APHH-Beijing programme. A total of seven responses was received. Despite constituting a relatively small group, the participants Those who participated in the elicitation-included researchers with expertise in compiling emissions inventory for the region of interest and researchers who used the same a priori emissions inventory in their own work. The fact that their responses were largely consistent also backs the credibility of the results. Specifically, the participants For each emissions parameter, they were invited to advise a lower and an upper bound of uncertainty for each emission parameter, such that it would be very unlikely for the true value to fall outside this range. The responses from the first round of elicitation were sent back to the participants anonymously for review. Finally, the maximum and minimum values advised by all participants for each parameter in the second round were adopted (Table 1, column Initial PEE). These wide uncertainty ranges also compensated for the small size of the expert group.

-line 208/209: Although in the simulation CO is treated as inert species, in general, it does affect the NOx concentrations via O3 chemistry (Gaubert et al., 2020, Correcting model biases of CO in East Asia: impact on oxidant distributions during KORUS-AQ, Atmos. Chem. Phys., 20, 14617–14647, https://doi.org/10.5194/acp-20-14617-2020). How does this assumption influence the emission estimation? What is the benefit from adding CO perturbations to the parameter field if CO is treated as inert? Both emitted species (NOx, CO) could as well be separately optimized.

This study was aimed at improving emission estimates for both CO and NOx. For efficiency, the uncertainty ranges of emission parameters for CO and NOx were elicited at the same time. We agree with the reviewer that since CO is treated as inert in ADMS-Urban, CO and NOx could be optimised separately, in which case two 70-member PEE would be required (following the rule of thumb of ten times the number of perturbed parameters). By perturbing all CO and NOx parameters simultaneously, the total number of model simulations performed is unchanged, yet the sampling density is doubled.

We also agree with the reviewer that on regional to global scales, the chemical loss of CO via reaction with OH is an important source of O3. However, our study area is 105 km  $\times$  144 km. With an annual mean wind speed of 2.8 m/s (in 2016), it would only take 17.6 hours for CO emitted from one corner of the domain to be transported diagonally to the other corner and subsequently out of the domain. This is much shorter compared to the lifetime of CO of 1 to 2.5 months. Therefore, we think that the chemical sink of CO should not have a measurable

effect on  $O_3$  and  $NO_2$  concentrations at this spatial scale. The effect of the assumption of CO being inert on the derived  $NO_x$  emission estimates should be minimal.

- line 210-213: Errors may not be the total emissions but the spatial distribution of the emissions, which is not addressed with the 14 parameter setup of the analysis. A discussion is appreciated on how this influences the results (especially locally close to emissions sources).

We thank the reviewer for this comment. We think in the prior emission estimates, biases in the spatial distribution of emissions co-exist with biases in the magnitude of emissions, as suggested by Fig. 4. In other words, the base emissions are positively biased in most areas, but more so in the central areas. We acknowledge that given the design of the adjusted PEE, biases in the spatial distribution of emissions could be propagated into the *a posteriori* emission estimates. As mentioned in our response to the reviewer's 3rd comment, such biases would be of greater concern if the *a posteriori* estimates were to be used in a gridded format. We have included this issue in the discussion of uncertainties:

Two types of uncertainties Uncertainties may be associated with the base emissions are twofold...mainly driven by spatial proxies that over-allocated industrial emissions to urban areas. Such uncertainties may have contributed to the spatial inhomogeneities of the biases in the base emissions revealed in Fig. 4 and, to a certain extent, propagated into the derived emission estimates, as spatially uniform perturbations were applied when constructing the adjusted PEE. Hence, biases in the spatial distribution of emissions may also be present in the *a posteriori* estimates, despite improvement in terms of the total magnitude. In comparison, the propagation of inherent uncertainties in the base emissions is of less concern. Though most emission parameters were defined relative to the corresponding values in the base emissions for an efficient perturbation, their uncertainty ranges were ultimately constrained solely by the observations.

- line 233: "high resolution" is a rather open statement. Emissions data are available at 3 km resolution. It would be good to add the exact resolution (horizontally and vertically).

We thank the reviewer for this comment. As explained in response to the reviewer's 7th comment regarding simulation setup, we only used ADMS-Urban to output pollutant concentrations at measurement locations, and thus did not mention an output resolution. Here the "high resolution" was referred to the possible output resolution of ADMS-Urban, which is not limited by the resolution and format of the input emissions. This is because the model is based on the analytical Gaussian plume formula (and its derivations). If a large number of point locations were defined, the model can produce fine concentration contours (e.g. Biggart et al., 2020) based on the exact distance of these points to emission sources. When outputting in a gridded format, the output resolution can also be finer than that of the gridded emissions.

- line 240-242: How is the city defined in the model? Are buildings represented as domain

**boundaries? If not, how is the local street canyon flow represented (e. g. channeling, overflow, small scale vorticies)? I expect from the manuscript that the model is not a LES model?**

The reviewer is correct that ADMS-Urban is not a large-eddy simulation model. It can model the effects of buildings and street canyons (albeit in a simplified manner compared to LES models) when provided with additional input data including the height, width and angle of buildings, the length, height and asymmetry of street canyons, etc. However, these data were unavailable for this study. Thus, local disturbance to the large-scale meteorological field is only accounted for by a roughness length parameter and a minimum Obukhov length parameter, which are set to recommended values in previous literature (e.g. Stewart and Oke, 2012), as described in detail in our previous paper (Yuan et al., 2021). The lines are modified to read:

...archived in the NOAA Integrated Surface Database (Smith et al., 2011). Local disturbance to the mean flow field by individual buildings and street canyons was not accounted for as such data were unavailable. Nonetheless, To account for differences in the local near-surface dynamics between at the weather observatory (situated in open landscape) and at the measurement sites (the majority of which are located in built-up areas), were represented by different values of roughness lengths and minimum Obukhov lengths, as described in Yuan et al. (2021).

-line 325: change "length" to "number" Done.

- line 332 – 335: I feel the reasoning in this statement is not correct. The fact that there is a large spread in the MSE depending on total NOx emissions does not necessarily mean that the emissions are higher and overestimated to a larger extent. It is rather the distribution of the MSE depending on total NOx emissions that lead to this conclusion (rapid increase in MSE for lower total NOx emissions and constant increase of MSE with increasing total NOx emissions).

**- line 335 – 338: I don't really understand this reasoning. Please rephrase.**

We thank the reviewer for these two comments. In this paragraph, we drew two conclusions from Fig. 4a. Firstly, with increasing input NOx emissions, the MSE in hourly NO2 concentrations at also increases continuously. This applies to all sites (including YLD where it is not evident from the figure as all MSEs fall within the 1st quartile). As rightly pointed out by the reviewer, this is an indication that NOx emissions are overestimated, both in the base emissions and in most members of the adjusted PEE. If NOx emissions were underestimated in many members of the PEE, the MSEs would first decrease to a point when the absolute bias in the emissions was the smallest. Secondly, the increase in MSE is not uniform across different sites. At most urban and traffic monitoring sites, the MSEs rapidly increase from very low values (i.e. within the 1st quartile) to very high values (i.e. within the 4th quartile), while the MSEs at other site types span a narrower range. We believe that this is due to higher base emissions (and effectively more pronounced overestimation) at these locations, for the simple reason that the same set of scaling factors would result in a wider range of scaled values, when applied to a larger base value. We have rephrased this paragraph to read:

...Figure 4a shows a distinct trend of increasing MSEs with growing annual total NOx emissions, such that at most sites, the base run with input emissions at the upper end of the scale (see Fig. 2a) is outperformed by most of the PEE simulations. Although a single MSE does not differentiate between over- and underestimation, this clear positive association between MSEs and NOx emissions suggests that NOx emissions are a-positively biased, both in the base emissions and in most members of the PEE. If NOX emissions were negatively biased in a considerable subset of the ensemble members, the MSEs would first decrease as emissions increased, until the absolute bias in the emissions reached a minimum. It is also evident that the base run is generally associated with larger errors at urban and traffic monitoring sites compared to other sites. This is also seen in most individual PEE simulations. Moreover, the increase in MSEs with increasing emissions is more rapid at these locations, resulting in a wider range of errors associated with the ensemble of simulations typically span a wider range at these locations. This is an indication that the base emissions are larger in magnitude of emissions in the central areas (where these sites are situated, see Fig. 1) are higher than those in the periphery and are overestimated to a larger extent-in the base emissions. Though spatially uniform scaling factors were applied within the study area, regions with a higher value in the base emissions would show larger variations in the perturbed emissions and thus model errors (as a result of being larger in magnitude)....

- discussion on Fig. 4: A discussion on the fact that some stations show almost no sensitivity to the underlying  $NO_X$  emissions is appreciated. How about the impact of other emissions (e. g. VOC, CO) on the  $O_3$  concentration. As stated in line 349,  $NO_X$  does not seem to be the only limiting factor for  $O_3$  concentrations. A discussion on further improvements would be nice.

We thank the reviewer for this comment. We have attached an alternate version of Fig. 4 below, which shows the variations in MSE as a function of the input annual total NOx emissions in more detail. (In comparison, Fig. 4 shows more clearly which MSEs are associated with a specific simulation). Panel (a) shows that the hourly NO2 concentrations at all sites are in fact sensitive to the input NOx emissions, with growing MSEs with increasing emissions. The sensitivity is smaller at most suburban, clean and regional background sites (explained in our response to the previous comment), yet the MSEs still increase with higher emissions. An example is the site YLD, where the small sensitivity is concealed by the four-tiered colour scale in Fig. 4a but is evident here.

Panel (b) also shows that at most sites, the MDA8 O3 concentrations are also sensitive to the input NOx emissions. However, several exceptions stand out. At the sites DSH and MTG, the MSEs do not appear to be associated with the emissions. At the sites HR and LLH, the MSEs even decrease with higher emissions. As the reviewer rightly pointed out, this suggests that there are other important limiting factors for the MDA8 O3 concentrations. This is also reflected in the low coefficients of determination calculated for linear functions of (log transformed) O3 as a function of NOx (see Fig. S9b in the revised Supplementary Information). As mentioned above in our response to the reviewer's comment on line 208/209 (in the original manuscript), we believe the effects of CO concentrations are negligible within the

study area. For a discussion on the impact of VOC concentrations, the reviewer is kindly referred to our response to the reviewer's 4th comment.

Virgin virg

 $(\mu g m^{-3})^2$

(b) MSE in MDA8 O3 concentration

MSE in hourly NO2 concentration

(a)

---

## Author Response (AR1)

We are very grateful to both reviewers for their constructive feedback. We have revised the manuscript and hope that the revisions address the reviewers' comments. In the following, each comment and our corresponding response are sequentially listed, in **bold** and plain text, respectively. Text in the original manuscript is shown in red with  marking deletions. New text in the revised manuscript is coloured blue. We will be submitting two versions of the revised manuscript: a clean version and a version with tracked changes.

**Anonymous Referee #1**

**- Title: I am not sure what the novelty in the perturbed emission ensemble is. Thus, I suggest to remove "novel" from the title.**
We thank the reviewer for this comment. Also taking the suggestion from Referee #2, the title now reads:

Improving $NO_X$ emission estimates in Beijing using network observations and a  perturbed emissions ensemble

**- The evaluation of the ensemble members is mainly based on the MSE. However, I was wondering if the ensemble shows a sign-change in the bias of $NO_2$ concentrations, which would further support the estimation of the optimal emission data.**
We agree with the reviewer that the MSE does not indicate the sign of model biases. Thus, we used the normalised mean bias factor (NMBF) in the preliminary evaluation of both the initial and the adjusted PEE simulations. The results are now presented in Fig. S2 in the revised Supplementary Information (whereby panel (a) corresponds to Fig. S1 in the original Supplementary Information) and are also shown below. Panel (a) reveals that at 22 long-term monitoring sites, the initial PEE simulations fail to output $NO_2$ concentrations comparable to the observations. Specifically, $NO_2$ concentrations at 19 sites are overestimated by the entire ensemble, while those at 3 other sites are underestimated. This widespread positive bias prompted us to decrease the lower bounds of uncertainty (and thus widen the uncertainty ranges) for most emission parameters. As shown in panel (b), the adjusted PEE simulations are associated with larger ranges of NMBFs. At 3 more sites, the range of NMBFs crosses the dashed line representing zero bias, indicating that more $NO_2$ concentrations comparable to the observations are simulated with the adjusted PEE.

NMBF in annual mean NO$_2$ concentrations

[Figure]

Initial PEE simulations      Adjusted PEE simulations

×   Base emissions simulation      ◇   Base run

Figure S1S2. Distribution of the normalised mean bias factors (NMBF) in annual mean NO$_2$ concentrations associated with (a) the initial and (b) the adjusted perturbed emissions ensemble (PEE) simulations and the simulation forced with the base emissions at each long-term monitoring site. In each panel, the simulation forced with the base emissions is also shown. Note that as different background levels of NO$_2$ and O$_3$ are input (in accordance with the initial and the adjusted PEE simulations), these two simulations are not identical, also indicated by the different NMBFs. The monitoring sites are colour-coded according to the site type: urban site (magenta), traffic monitoring site (purple), suburban site (orange), clean site (light green) and regional background site (green). The circle at the leftmost end of the boxplot for YLD in panel (b) represents an outlying PEE simulation (i.e. with a NMBF outside 1.5 times the interquartile range below the lower quartile).

**- I understand that the choice of spatiotemporal uniform emission perturbations suggests an evaluation of averaged concentrations over all stations. An evaluation at single stations was initiated by e. g. Fig. 5, but I would have expected a more detailed investigation of the ensemble members in different regions. Potentially, the results allow also for a spatially heterogeneous emission correction.**

We thank the reviewer for this comment. As our conclusions were drawn with respect to annual emissions within the modelling domain, we think an evaluation of the PEE simulations' average performance across all measurement sites is appropriate. We agree with the reviewer

in that Fig. 4 and Fig. 5 reveal disparities in the simulations' performance at individual sites, suggesting the presence of spatial biases in the *a posteriori* emission estimates derived from these simulations. If gridded *a posteriori* emission estimates were to be derived, spatially heterogeneous perturbations to the prior emissions would be ideal, and the evaluation of the simulations would be best performed locally at individual sites within certain localisation scale. We think these are beyond the scope of this study but are important directions for future development of our method.

**- Although VOC emissions (and background concentrations) are included in the model, the impact on these emissions and potential uncertainties is not addressed adequately. Especially in high NO$_X$ concentrations, the O$_3$ concentrations depend highly on the available VOC. The manuscript only considers NO$_X$ emissions as main source of uncertainty. A discussion on the impact of this choice is appreciated.**

We agree with the reviewer that emissions and background concentrations of VOCs have an impact on the NO$_X$-O$_3$ chemistry and thus the *a posteriori* emission estimates of NO$_X$. Ideally, VOC emissions should be perturbed and constrained along with NO$_X$ emissions. This was not possible for two reasons. On one hand, VOC concentrations participate in the NO$_X$ photolytic chemistry scheme in ADMS-Urban as a source of radicals, but are not themselves affected by the chemical calculations. On the other hand, long-term network observations of VOC concentrations were unavailable. Hence, we discussed the impact of VOCs on the derived NO$_X$ emissions qualitatively using the two ROC concentration sensitivity simulations described in Section 4. This may not have been clear in the original manuscript. The relevant paragraph is revised as follows:

The effect of organic radicals on the partition of NO$_X$ between NO$_2$ and NO is shown in Fig. 8c by varying the concentrations of ROC. As explained above, ROC concentrations are controlled by both the TVOC concentrations (that result from the input emissions and background levels of VOC) and a reactivity coefficient which was set to 0.1 in R97 (as with other adjusted PEE simulations).  With fixed TVOC concentrations, using a coefficient of 0.2 doubles the ROC available to produce HO$_2$ and RO$_2$, leading  to an even more pronounced overestimation of NO$_2$, accompanied by an underestimation of NO. In contrast, halving the ROC concentrations by using a coefficient of 0.05 partitions less of the NO$_X$ emitted into NO$_2$. This highlights that the emissions and background concentrations of VOC (which are not evaluated in this study due to a lack of observations) also have an impact on the modelled NO$_X$ photolytic chemistry and thus the *a posteriori* emission estimates of NO$_X$. It is also worth noting that biogenic VOCs are likely underestimated in the current simulations, as these are only represented by one of the 8 species (i.e. isoprene) output by the CAMS reanalysis product used to approximate the background levels of TVOC, and are not represented at all in the base emissions (which include anthropogenic sources only). Despite low in concentrations in the study area (compared to anthropogenic VOCs) (Mo et al., 2018), they are associated with high radical production and thus O$_3$ creation potentials.

coefficient in ADMS-Urban version 5 is thus supported. Unlike background pollutant concentrations of $NO_X$ and $O_3$, however, the effect of ROC concentrations VOCs on the modelled $NO_X$-O chemistry is restricted to daylight hours, as they only produce radicals in the presence of solar radiation in the model.

**- A discussion on the representativity of observation sites (especially urban and traffic) is required. Is the model resolution sufficient to be compared the traffic measurement stations?**
We thank the reviewer for this comment. We agree with the reviewer that the gridded emissions dilute traffic emissions within each grid cell, which could result in an underestimation of roadside $NO_2$ concentrations. In fact, this was found at some sites in Beijing in ADMS-Urban simulations input with another emissions inventory of the same 3 km × 3 km resolution (Biggart et al., 2020). However, as our conclusions were drawn with respect to annual traffic emissions within the domain using measurements of all 33 sites as constraints, we believe this is a minor issue, since potential underestimation at traffic monitoring sites would be accompanied by overestimation at other sites at some distance. In comparison, we think the under-representativeness of the existing site network for the power and industrial emissions is a major limitation. We have included a short discussion on this in Section 4:
Meanwhile, uncertainties Uncertainties in the observational constraints are twofold. While those due to measurement errors are mostly likely small, which is as demonstrated by the consistency in the results derived using two independent sets of observations, the underrepresentation of the existing observations of power and industrial sources prohibited an update of emission strengths from these sources.

**- the wording initial PEE and optimized PEE is somehow misleading. Only as I have finished section 2 I have understood that this approach is not a data assimilation or inversion method. Maybe "adjusted PEE" instead of "optimized PEE" would be clearer. In the context of observations, "optimized" always feels like there is some optimization method applied, which is certainly not the case in this manuscript.**
We thank the reviewer for this suggestion. The term "adjusted PEE" is now used instead of "optimised PEE" throughout the revised manuscript.

**- To the simulation setup: It is not really clear, which simulations have been done. There is a base run with additional 140 member ensemble with perturbed $NO_X$ emissions. However, the simulation episode should be state here explicitly (am I right that the full year 2016 was simulated?), also the model resolution (horizontal and vertical) is missing. A link to the discussion section, where the limitations introduced by model simplifications is discussed, would be good. It would have been easier for the understanding of the results that only the optimized PEE is used for the simulations.**
We thank the reviewer for these comments. We did not mention exact spatial resolutions of the simulations, as we did not output pollutant concentrations in a gridded format, but only those at measurement locations. More explanations on the possible output resolution of the

model are provided below in our response to the reviewer's comment on line 233 (in the original manuscript). The total number and the time frame of the simulations performed are added in the last paragraph under Section 2.3, where it is also made clear that only the adjusted PEE simulations (and not the initial PEE simulations) were used to derive *a posteriori* emission estimates:

The input meteorology data and background pollutant concentrations described above provided the same lateral boundary conditions for the 140 adjusted PEE simulations, among which only the emissions of $NO_X$ (and $NO_2$) varied. An additional simulation forced with these boundary conditions and the base emissions was also performed and is hereinafter referred to as the base run. All 141 simulations were run for the whole year of 2016 to produce hourly pollutant concentrations at each measurement location (see Fig. 1). Output of these simulations were then compared to measurements to derive *a posteriori* emission estimates.

**- line 52: A discussion on the local anthropogenic and biogenic share of $NO_X$ (and esp. VOC) emissions would be appreciated. Are biogenic emissions in this regions (especially in summer) negligible compared to the anthropogenic emissions?**

We thank the reviewer for this valuable input. We have revised this paragraph to include a short discussion on the relative importance of anthropogenic and soil $NO_X$ emissions in China, as the latter represents a larger natural source compared to lightning (Lin, 2012).

 $NO_X$ can be produced from both anthropogenic and natural/biogenic sources such as fossil fuel combustion, biomass burning, soil microbial processes and lightning Lee et al., 1997). Global total anthropogenic $NO_X$ emissions flattened around 2008, as reductions in Europe and North America were offset by increases in Asia (Hoesly et al., 2018). China, in particular, witnessed a rapid rise in anthropogenic $NO_X$ emissions until 2011-2012…stricter vehicle emission standards combined with accelerated fleet turnover (Liu et al., 2020). Decreases in anthropogenic sources are accompanied by an increased importance of soil $NO_X$ emissions, which are largely driven by nitrogen fertiliser application and can reach up to 20% of the anthropogenic emissions in the crop growing season in some regions with high agricultural activities (Lu et al., 2021). These emissions are relatively poorly quantified and currently unabated (State Council of the People's Republic of China, 2018).

However, for our study area centred around Beijing, and on an annual scale (as are the derived emission estimates), we believe that soil $NO_X$ emissions are negligible compared to anthropogenic $NO_X$ emissions. This is also supported by the Supplementary Fig.S1 in Lu et al. (2021) which shows Beijing dominated by "high anthropogenic $NO_X$ emission" model grids. Hence, soil $NO_X$ emissions are not discussed further in the revised manuscript.

For a discussion on the impact of VOCs, the reviewer is kindly referred to our response to the reviewer's 4[th] comment, where we have also included a discussion on the likely underrepresentation of biogenic VOCs in the simulations.

**- Line 56-58: Citations for the different action plans required**

References to the action plans are added as:

Emission reduction targets were first announced in the 12th Five-Year Plan (2011-2015) (People's Republic of China, 2011), followed by the Action Plan on Prevention and Control of Air Pollution (2013-2017) (State Council of the People's Republic of China, 2013) and the Three-Year Action Plan for Winning the Blue Sky Defence Battle (2018-2020) (State Council of the People's Republic of China, 2018).

**-line 64: ...method, which... (add a comma)**

The sentence is modified to read:

Some have used a bottom-up method  that combines specific emission factors (i.e. mass of a pollutant emitted per unit fuel consumption or industrial production)  with the corresponding  activity rates (i.e. fuel consumption or industrial production),…

**-line 65/66: I guess you are talking about the amount of studies investigating emission data, please be more precise: which data? What is the large amount of the data? How can data solve the time-lag issue?**

Here we were referring to the large amount of input data (e.g. emission factors and energy consumption data) required to establish a bottom-up emissions inventory, most of which are not available real-time. Thus, the collection and compilation of these data result in a considerable time-lag in the occurrence of emissions inventories, typically one to several years. The text is modified as follows to avoid ambiguity:

However,  the underlying data are mostly not immediately available,  resulting in an inevitable time-lag between the occurrence of emissions and the establishment of an inventory (Janssens-Maenhout et al., 2015).

**- line 68: It may be worth elaborate on emission uncertainties and their impact**

We thank the reviewer for this comment. A reference for the uncertainty estimates for MEIC is provided in the 6[th] paragraph under the discussion section. Two references for the impact of emission uncertainties are added here:

…large and poorly quantified uncertainties into the emission estimates (Hong et al., 2017; Zhao et al., 2011), which can be further propagated through modelled pollutant concentrations into disease or mortality burden (Crippa et al., 2019) and economic loss estimates (Solazzo et al., 2018).

**- It would be worth elaborate more about the pros and cons of the different methods you are summarizing in the introduction. Why are you proposing the new method, what is the strength of your method compared to the other methods?**

We thank the reviewer for this comment. We have amended this paragraph such that our discussion of the pros and cons of the bottom-up and top-down methods stands out more clearly. The strength of our method is summarised in a sentence at the start of the next paragraph:

Some have used a bottom-up method  that combines specific emission factors (i.e. mass of a pollutant emitted per unit fuel consumption or industrial production)  with the corresponding  activity rates (i.e. fuel consumption or industrial production), thus providing sector- or process-resolved emission estimates (Liu et al., 2016; Zhang et al., 2009; Zhao et al., 2013; Zheng et al., 2018). However,  the underlying data are mostly not immediately available,  resulting in an inevitable time-lag between the occurrence of emissions and the establishment of an inventory (Janssens-Maenhout et al., 2015). Moreover,  they can introduce potentially large and poorly quantified uncertainties into the emission estimates (Hong et al., 2017; Zhao et al., 2011), which can be further propagated through modelled pollutant concentrations into disease or mortality burden (Crippa et al., 2019) and economic loss estimates (Solazzo et al., 2018). Other studies have inferred top-down estimates of emissions using satellite observations ...four-dimensional variational assimilation (Kurokawa et al., 2009) have been increasingly adopted to combine satellite observations and CTM simulations with prior emission estimates to derive *a posteriori* emission estimates. These inverse methods  provide more timely emission estimates of high spatial and temporal coverage (based on the nature of satellite observations). Nonetheless, the derived emission estimates are not resolved by source sector. They are also subject to uncertainties propagated from the satellite retrievals and the model simulations...

This study introduces a novel approach that provides timely updates of a priori emission estimates by source sector using readily available in-situ air quality observations. Using this approach,  *a priori* $NO_x$ emissions in a bottom-up inventory compiled for Beijing for the year 2013  are updated for 2016 .

**-line 83 – 86: This statement is not only valid for satellite data. Insufficient chemistry always influences the model results and, thus, the analysis.**

We agree with the reviewer that any analysis involving model simulations can be affected by insufficient chemistry pathways in the model. Here we were referring specifically to the effect of model chemistry on the inversely derived emission estimates. The text is revised to avoid ambiguity:

For instance, Archer-Nicholls et al. (2021) showed large differences in the $NO_2$ column density simulated by two chemical mechanisms with different treatment of non-methane volatile organic compounds (NMVOCs), which are integrated into the same model with identical $NO_x$ emissions

. When used in inverse modelling, these modelled $NO_2$ quantities would result in different *a posteriori* $NO_X$ emissions.

**-line 106 -111: I feel like this is too much detail for the manuscript. Is it necessary to follow the study to know the accuracy of the measurement instruments?**

**- line 116-118: Also, is this information necessary for the manuscript? I don't feel so. Are the low-cost sensors influenced by a systematic error (bias) that may have an influence on the comparison?**

We thank the reviewer for these two comments. The low-cost sensors were co-located with reference instruments prior to deployment and a systematic bias was not found. The fact that the *a posteriori* emission estimates derived independently using reference instrument measurements and low-cost sensor measurements are consistent also indicates minimal influence of measurement errors. These lines are now deleted.

**- Table 1: Please include the night time definition for the initial and optimized ensemble in the caption. Also, the caption states night time fractions are in %, but values show ratios, please revise.**

We thank the reviewer for this comment and pointing out the error. The definitions for the night-time fraction of transport sector $NO_X$ emissions in the two PEEs are included in a footnote, and the respective minimum and maximum values adopted are corrected to read:

Table 1. Emission parameters[a] and the respective uncertainty ranges sampled by the initial and the  adjusted perturbed emissions ensembles (PEEs).

| Parameter | Initial PEE | |  Adjusted PEE | |
|---|---|---|---|---|
| | Min | Max | Min | Max |
| Industry sector ground level $NO_X$ emissions | 0.4 | 1.6 | 0.05 | 1.6 |
| Industry sector elevated $NO_X$ emissions | 0.4 | 1.4 | 0.05 | 1.4 |
| Power sector $NO_X$ emissions[b] | 0.2 | 1.4 | NA | NA |
| Power sector $NO_X$ emissions below 152 m | NA | NA | 0.05 | 1.6 |
| Power sector $NO_X$ emissions above 152 m | NA | NA | 0.05 | 1.6 |
| Residential sector $NO_X$ emissions | 0.4 | 1.5 | 0.05 | 1.5 |
| Transport sector $NO_X$ emissions | 0.4 | 2 | 0.05 | 1.5 |
| Night-time fraction of transport sector $NO_X$ emissions[c] |  10 |  40 |  10 |  30 |

[a] All parameters are defined as ratios of the 2016 emissions to the base emissions from 2013, except for the night-time fraction of transport sector $NO_X$ emissions which is defined as a percentage (%) of the daily totals in 2016.

[b] Power sector $NO_X$ emissions are effectively represented by one parameter in the initial PEE. In the  adjusted PEE, the emissions are split into two parameters, namely emissions below and above 152 m.

[c] Night-time fraction of transport sector $NO_X$ emissions is defined as those occurring during

11am-6am in the initial PEE. In the adjusted PEE, it is modified to $NO_X$ emitted during 0-5 am from the transport sector.

**-Line 178-179: Please include the number of experts that contributed to the poll.**
We thank the reviewer for this comment. More information about the expert elicitation is included:

An online questionnaire was designed for the elicitation (available at https://cambridge.eu.qualtrics.com/jfe/form/SV_3eGxf9XvC7WXESV, last accessed 14 April 2022) and circulated via the mailing list of the APHH-Beijing programme. A total of seven responses was received. Despite constituting a relatively small group, the participants  included researchers with expertise in compiling emissions inventory for the region of interest and researchers who used the same *a priori* emissions inventory in their own work. The fact that their responses were largely consistent also backs the credibility of the results. Specifically, the participants  were invited to advise a lower and an upper bound of uncertainty for each emission parameter, such that it would be very unlikely for the true value to fall outside this range. The responses from the first round of elicitation were sent back to the participants anonymously for review. Finally, the maximum and minimum values advised by all participants for each parameter in the second round were adopted (Table 1, column *Initial PEE*). These wide uncertainty ranges also compensated for the small size of the expert group.

**-line 208/209: Although in the simulation CO is treated as inert species, in general, it does affect the $NO_X$ concentrations via $O_3$ chemistry (Gaubert et al., 2020, Correcting model biases of CO in East Asia: impact on oxidant distributions during KORUS-AQ, Atmos. Chem. Phys., 20, 14617–14647, https://doi.org/10.5194/acp-20-14617-2020). How does this assumption influence the emission estimation? What is the benefit from adding CO perturbations to the parameter field if CO is treated as inert? Both emitted species ($NO_X$, CO) could as well be separately optimized.**
This study was aimed at improving emission estimates for both CO and $NO_X$. For efficiency, the uncertainty ranges of emission parameters for CO and $NO_X$ were elicited at the same time. We agree with the reviewer that since CO is treated as inert in ADMS-Urban, CO and $NO_X$ could be optimised separately, in which case two 70-member PEE would be required (following the rule of thumb of ten times the number of perturbed parameters). By perturbing all CO and $NO_X$ parameters simultaneously, the total number of model simulations performed is unchanged, yet the sampling density is doubled.

We also agree with the reviewer that on regional to global scales, the chemical loss of CO via reaction with OH is an important source of $O_3$. However, our study area is 105 km $\times$ 144 km. With an annual mean wind speed of 2.8 m/s (in 2016), it would only take 17.6 hours for CO emitted from one corner of the domain to be transported diagonally to the other corner and subsequently out of the domain. This is much shorter compared to the lifetime of CO of 1 to 2.5 months. Therefore, we think that the chemical sink of CO should not have a measurable

effect on $O_3$ and $NO_2$ concentrations at this spatial scale. The effect of the assumption of CO being inert on the derived $NO_X$ emission estimates should be minimal.

**- line 210-213: Errors may not be the total emissions but the spatial distribution of the emissions, which is not addressed with the 14 parameter setup of the analysis. A discussion is appreciated on how this influences the results (especially locally close to emissions sources).**

We thank the reviewer for this comment. We think in the prior emission estimates, biases in the spatial distribution of emissions co-exist with biases in the magnitude of emissions, as suggested by Fig. 4. In other words, the base emissions are positively biased in most areas, but more so in the central areas. We acknowledge that given the design of the adjusted PEE, biases in the spatial distribution of emissions could be propagated into the *a posteriori* emission estimates. As mentioned in our response to the reviewer's 3rd comment, such biases would be of greater concern if the *a posteriori* estimates were to be used in a gridded format. We have included this issue in the discussion of uncertainties:

Two types of uncertainties may be associated with the base emissions …mainly driven by spatial proxies that over-allocated industrial emissions to urban areas. Such uncertainties may have contributed to the spatial inhomogeneities of the biases in the base emissions revealed in Fig. 4 and, to a certain extent, propagated into the derived emission estimates, as spatially uniform perturbations were applied when constructing the adjusted PEE. Hence, biases in the spatial distribution of emissions may also be present in the *a posteriori* estimates, despite improvement in terms of the total magnitude. In comparison, the propagation of inherent uncertainties in the base emissions is of less concern. Though most emission parameters were defined relative to the corresponding values in the base emissions for an efficient perturbation, their uncertainty ranges were ultimately constrained solely by the observations.

**- line 233: "high resolution" is a rather open statement. Emissions data are available at 3 km resolution. It would be good to add the exact resolution (horizontally and vertically).**

We thank the reviewer for this comment. As explained in response to the reviewer's 7th comment regarding simulation setup, we only used ADMS-Urban to output pollutant concentrations at measurement locations, and thus did not mention an output resolution. Here the "high resolution" was referred to the possible output resolution of ADMS-Urban, which is not limited by the resolution and format of the input emissions. This is because the model is based on the analytical Gaussian plume formula (and its derivations). If a large number of point locations were defined, the model can produce fine concentration contours (e.g. Biggart et al., 2020) based on the exact distance of these points to emission sources. When outputting in a gridded format, the output resolution can also be finer than that of the gridded emissions.

**- line 240-242: How is the city defined in the model? Are buildings represented as domain**

**boundaries? If not, how is the local street canyon flow represented (e. g. channeling, overflow, small scale vorticies)? I expect from the manuscript that the model is not a LES model?**

The reviewer is correct that ADMS-Urban is not a large-eddy simulation model. It can model the effects of buildings and street canyons (albeit in a simplified manner compared to LES models) when provided with additional input data including the height, width and angle of buildings, the length, height and asymmetry of street canyons, etc. However, these data were unavailable for this study. Thus, local disturbance to the large-scale meteorological field is only accounted for by a roughness length parameter and a minimum Obukhov length parameter, which are set to recommended values in previous literature (e.g. Stewart and Oke, 2012), as described in detail in our previous paper (Yuan et al., 2021). The lines are modified to read:

…archived in the NOAA Integrated Surface Database (Smith et al., 2011). Local disturbance to the mean flow field by individual buildings and street canyons was not accounted for as such data were unavailable. Nonetheless,  differences in the  near-surface dynamics  at the weather observatory (situated in open landscape) and at the measurement sites (the majority of which are located in built-up areas) were represented by different values of roughness length and minimum Obukhov length, as described in Yuan et al. (2021).

**-line 325: change "length" to "number"**

Done.

**- line 332 – 335: I feel the reasoning in this statement is not correct. The fact that there is a large spread in the MSE depending on total NO$_X$ emissions does not necessarily mean that the emissions are higher and overestimated to a larger extent. It is rather the distribution of the MSE depending on total NO$_X$ emissions that lead to this conclusion (rapid increase in MSE for lower total NO$_X$ emissions and constant increase of MSE with increasing total NOx emissions).**

**- line 335 – 338: I don't really understand this reasoning. Please rephrase.**

We thank the reviewer for these two comments. In this paragraph, we drew two conclusions from Fig. 4a. Firstly, with increasing input NO$_X$ emissions, the MSE in hourly NO$_2$ concentrations at also increases continuously. This applies to all sites (including YLD where it is not evident from the figure as all MSEs fall within the 1$^{st}$ quartile). As rightly pointed out by the reviewer, this is an indication that NO$_X$ emissions are overestimated, both in the base emissions and in most members of the adjusted PEE. If NO$_X$ emissions were underestimated in many members of the PEE, the MSEs would first decrease to a point when the absolute bias in the emissions was the smallest. Secondly, the increase in MSE is not uniform across different sites. At most urban and traffic monitoring sites, the MSEs rapidly increase from very low values (i.e. within the 1$^{st}$ quartile) to very high values (i.e. within the 4$^{th}$ quartile), while the MSEs at other site types span a narrower range. We believe that this is due to higher base emissions (and effectively more pronounced overestimation) at these locations, for the simple reason that the same set of scaling factors would result in a wider range of scaled values, when applied to a larger base value. We have rephrased this paragraph to read:

…Figure 4a shows a distinct trend of increasing MSEs with growing annual total $NO_X$ emissions, such that at most sites, the base run with input emissions at the upper end of the scale (see Fig. 2a) is outperformed by most of the PEE simulations. Although a single MSE does not differentiate between over- and underestimation, this clear positive association between MSEs and $NO_X$ emissions suggests that $NO_X$ emissions are  positively biased, both in the base emissions and in most members of the PEE. If $NO_X$ emissions were negatively biased in a considerable subset of the ensemble members, the MSEs would first decrease as emissions increased, until the absolute bias in the emissions reached a minimum. It is also evident that the base run is generally associated with larger errors at urban and traffic monitoring sites compared to other sites.  Moreover, the increase in MSEs with increasing emissions is more rapid at these locations, resulting in a wider range of errors associated with the ensemble of simulations . This is an indication that the base emissions are larger in magnitude  in the central areas (where these sites are situated, see Fig. 1)  than  in the periphery and are overestimated to a larger extent . Though spatially uniform scaling factors were applied within the study area, regions with  higher  base emissions would show larger variations in the perturbed emissions and thus model errors ….

**- discussion on Fig. 4: A discussion on the fact that some stations show almost no sensitivity to the underlying $NO_X$ emissions is appreciated. How about the impact of other emissions (e. g. VOC, CO) on the $O_3$ concentration. As stated in line 349, $NO_X$ does not seem to be the only limiting factor for $O_3$ concentrations. A discussion on further improvements would be nice.**

We thank the reviewer for this comment. We have attached an alternate version of Fig. 4 below, which shows the variations in MSE as a function of the input annual total $NO_X$ emissions in more detail. (In comparison, Fig. 4 shows more clearly which MSEs are associated with a specific simulation). Panel (a) shows that the hourly $NO_2$ concentrations at all sites are in fact sensitive to the input $NO_X$ emissions, with growing MSEs with increasing emissions. The sensitivity is smaller at most suburban, clean and regional background sites (explained in our response to the previous comment), yet the MSEs still increase with higher emissions. An example is the site YLD, where the small sensitivity is concealed by the four-tiered colour scale in Fig. 4a but is evident here.

Panel (b) also shows that at most sites, the MDA8 $O_3$ concentrations are also sensitive to the input $NO_X$ emissions. However, several exceptions stand out. At the sites DSH and MTG, the MSEs do not appear to be associated with the emissions. At the sites HR and LLH, the MSEs even decrease with higher emissions. As the reviewer rightly pointed out, this suggests that there are other important limiting factors for the MDA8 $O_3$ concentrations. This is also reflected in the low coefficients of determination calculated for linear functions of (log transformed) $O_3$ as a function of $NO_X$ (see Fig. S9b in the revised Supplementary Information). As mentioned above in our response to the reviewer's comment on line 208/209 (in the original manuscript), we believe the effects of CO concentrations are negligible within the

study area. For a discussion on the impact of VOC concentrations, the reviewer is kindly referred to our response to the reviewer's 4th comment.

**(a)**      MSE in hourly NO₂ concentration

[Figure]

**(b)**      MSE in MDA8 O₃ concentration

[Figure]

Figure R1. Mean square errors (MSE) in (a) hourly NO₂ concentrations and (b) daily maximum 8-hour mean (MDA8) O₃ concentrations associated with the adjusted perturbed emissions ensemble (PEE) simulations (circles colour-coded according to the input annual total NOₓ emissions with darker colour indicating higher values) and the base run (black circles) at each long-term monitoring site. The monitoring sites are arranged and colour-coded according to the site type: urban site (magenta), traffic monitoring site (purple), suburban site (orange), clean site (light green) and regional background site (green).

**- line 384 – 385: I don't really understand this. There is a change in MSE of O₃ with changing total NOₓ emissions in Fig. 4b. Here you state, that this is associated with the mMSE. Maybe**

**you can give examples how the mMSE is influenced (e. g. via changes in VOC concentrations by altering the $NO_X$ emissions?). Also in the discussion on Fig. 6, there is a dependence of the mMSE on the $NO_X$ emissions visible, which needs to be related to a lower correlation coefficient. Thus, in my opinion the decomposition of the MSE is mainly influenced by the changing correlation, which shifts the contribution to either the second or third term of Eq. 2 if the bias is negligible. I would like to see this discussed further.**

We disagree with the reviewer in the interpretation of Fig. 6. We believe it shows no dependence of the mMSEs (in the median MSEs in both hourly $NO_2$ concentrations and MDA8 $O_3$ concentrations) on the input $NO_X$ emissions. The mMSEs vary between simulations, but their variations do not appear correlated with variations in the emissions. This is also consistent with the fact that the mMSE is, by definition, less dependent on external forcings of the model, as mentioned in the paragraph under Eq. (2):

The last term, by definition, represents the proportion of the observed variance unexplained by the model. It summarises all non-systematic errors, including the noise and inherent variability (e.g. due to turbulence closure) in the observations as well as errors arising from the linearisation of non-linear processes, and is referred to as the minimum achievable MSE (mMSE).

We think this explains why the MSEs in MDA8 $O_3$, most of which are dominated by the mMSE (Fig. 5b), shows an overall weaker or less robust association with the input $NO_X$ emissions, compared to the MSEs in hourly $NO_2$ (Fig. 4). The latter are mostly made up of the variance or the bias error (Fig. 5a), which shows stronger dependence on the model inputs.

We agree with the reviewer in that for a given pollutant (i.e. with the observed variability fixed), the higher the correlation (when positive), the smaller is the mMSE. The magnitude of the variance error is then closer to the squared difference between the modelled and the observed variability. However, in this paragraph, we were comparing the mMSEs associated with hourly $NO_2$ and MDA8 $O_3$, the differences of which are much smaller compared to the differences between their MSEs. This is because the variance in the observed $O_3$ is about 3 or 4 times the variance in observed $NO_2$. Despite a better correlation in $O_3$, the unexplained portion (i.e. $1-r^2$) of the observed variance in $O_3$, that is, the mMSE, is still comparable to that in $NO_2$ (see Fig. S4 in the revised Supplementary Information). We did not discuss the relative weights of the three terms in MSE further, as we used the total magnitude of MSE for the evaluation of the PEE simulations. The decomposition was helpful in identifying the sources of model error, in particular, whether or not the error was largely driven by the input emissions.

**- line 413-414: this is only valid for uniform perturbations across the domain. Please add this information to the sentence.**

We thank the reviewer for this comment. This sentence is revised to read:

With spatially uniform perturbations, it is likely that several different combinations of emission parameter values result in similar concentrations at a particular location

**- line 421-422: Please add reference(s).**

**- line 422: An introduction to Fig. 7 is missing**

The start of the paragraph is updated as follows:

On account of the analysis above, we only used observations of $NO_2$ to constrain $NO_X$ emissions, which is also in line with numerous top-down emission optimisation studies using satellite observations of column $NO_2$ (e.g. Lamsal et al. 2011; Martin et al. 2003; Napelenok et al. 2008; Qu et al. 2017). Figure 7 shows the average performance for hourly $NO_2$ of individual PEE simulations against the value set for each emission parameter in Table 1. Figure 7(f)  reveals a strong positive correlation between the median MSE in hourly $NO_2$ of a simulation and the  input transport sector $NO_X$ emissions

**- line 618-620: Comparing Fig. 8a and 8b the impact of changing the input is almost as large as the variety within the top 5 % PEE members. Thus, I feel the change of input concentrations would have also a large impact on the uncertainty of emission estimates, potentially leading to larger uncertainties in the emissions. Please add a discussion on this impact.**

We thank the reviewer for raising this point. We agree that the variations in the slopes among the background concentration sensitivity simulations are comparable to those among the top performing 5% of the simulations with varying emissions, suggesting a potentially large impact of the input background levels on the *a posteriori* emission estimates. This impact is currently difficult to quantify as there are yet no widely accepted definitions for uniform background levels of $NO_X$ and $O_3$ within a large urban area. Nonetheless, we think the background levels input in the sensitivity simulation S5 (the slopes of which show the largest departure from those of the best PEE simulation R97) are unlikely to be representative of the actual background levels. In particular, the $10^{th}$ percentile baseline concentration of $O_3$ is likely substantially low biased. We have added a short discussion here to read:

…It simply highlights the impact of the input background concentrations of reactive pollutants on the model outputs of relevant species (and thus on the emission estimates inferred on the basis of these model outputs), which can be comparable to the impact of varying the input $NO_X$ emissions (amongst the top performing 5% of the adjusted PEE simulations) shown in Fig. 8a. This calls for further research into appropriate definitions for background levels of $NO_X$ and $O_3$ within a vast and heterogeneous urban area like the modelling domain in this study. Also, it is worth noting that the modelled chemistry is also influenced by the input background levels of NO…

**Anonymous Referee #2**

**-I found the title a little ambiguous. Consider changing from 'Improving NOx emissions…' to 'Improving NOx emissions estimates…'**

We agree with the reviewer about the ambiguity. Also taking the suggestion from Referee #1, the title now reads:

Improving NOX emissions estimates in Beijing using network observations and a  perturbed emissions ensemble

**-Line 115 – what height were the SNAQ sensors deployed at?**

We thank the reviewer for raising this point. The text has been amended to include measurement height information:

The measurements were made with low-cost sensors also deployed in a variety of  near-surface locations in Beijing (with an average measurement height of 8 m) and are hereinafter referred to as SNAQ (Sensor Network for Air Quality) (Fig. 1 and Table S2).

**-More details should be provided about the 'elicitation of expert knowledge' process. How many people were consulted? How did you select experts? Did you design a questionnaire which was sent to people? If so, could you include a copy of this questionnaire in the supplementary information?**

We thank the reviewer for this comment. More information about the expert elicitation is included, along with a link to the online questionnaire used:

An online questionnaire was designed for the elicitation (available at https://cambridge.eu.qualtrics.com/jfe/form/SV_3eGxf9XvC7WXESV, last accessed 14 April 2022) and circulated via the mailing list of the APHH-Beijing programme. A total of seven responses were received. Despite constituting a relatively small group, the participants  included researchers with expertise in compiling emissions inventory for the region of interest and researchers who used the same *a priori* emissions inventory in their own work. The fact that their responses were largely consistent also backs the credibility of the results. Specifically, the participants  were invited to advise a lower and an upper bound of uncertainty for each emission parameter, such that it would be very unlikely for the true value to fall outside this range. The responses from the first round of elicitation were sent back to the participants anonymously for review. Finally, the maximum and minimum values advised by all participants for each parameter in the second round were adopted (Table 1, column *Initial PEE*). These wide uncertainty ranges also compensated for the small size of the expert group.

**-Lines 135-136. Does this imply all profiles are the same for all pollutants in the inventory? Or is there a different diurnal, monthly and vertical profile for each pollutant. Please make this clearer.**

Within each source sector, the same profiles are applied to all pollutants. To clarify, this sentence now reads:

 each source sector is associated with a specific set of diurnal, monthly and vertical variation profiles that is applied to emissions of all pollutants from the sector.

**-Line 139. The authors describe the area that the base emissions cover in the text, but it**

**would be helpful to visualise this with a figure. Could the authors include a map of the base emissions (total or by source sector) to show the overlap with monitoring sites? This could be overlayed in Fig. 1 or included as a new figure in the supplementary information.**

We thank the reviewer for this comment. Maps of annual $NO_X$ emissions by source sector in the base emissions are included as Fig. S1:

[Figure]

Fig. S1. Annual $NO_X$ emissions from each source sector and grid cell (of 3 km × 3 km resolution) in the base emissions. For the industry and power sectors, emissions from all vertical layers are aggregated. The administrative divisions of Beijing are shown by light grey outlines.

Also, the following sentence is added to the first paragraph under Section 2.2:
…, hereinafter referred to as the base emissions. Annual $NO_X$ emissions in this region are shown in Fig. S1 by source sector.

**-Lines 206-210. I didn't understand why CO was perturbed in the model if it is treated as inert and will not affect $NO_X$ concentrations? Please add some lines to clarify why this was done.**

This study was aimed at improving emission estimates for both CO and $NO_X$. For efficiency, uncertainty ranges of emission parameters for CO and $NO_X$ were elicited at the same time. Perturbations to these parameters were also done simultaneously, resulting in the 140-member adjusted PEE (referred to as the optimised PEE in the original manuscript). As the reviewer correctly pointed out, CO and $NO_X$ are not interactive in ADMS-Urban, which means that a simultaneous perturbation is reasonable. The improved emission estimates for CO are

presented separately in Yuan et al. (2021). The text is modified as follows to avoid confusion:

 As this study also sought to improve emission estimates of CO in the base emissions (the results of which presented in Yuan et al. (2021)), the uncertainty ranges of  relevant emission parameters were also elicited and modified in the same processes as the $NO_X$ emission parameters.  The 14 parameters in total (i.e. 7 for $NO_X$, 7 for CO) determined for the adjusted PEE constituted a 14-dimensional uncertain space, which was probed efficiently using the maximin Latin hypercube sampling, which maximises the minimum inter-sample distance (Johnson et al., 1990). A rule of thumb is to have a sample size 10 times the dimension (Loeppky et al., 2009). We drew 140 samples, effectively doubling the sample size generally required (i.e. if only $NO_X$ emission parameters were perturbed). A simultaneous perturbation to both CO and $NO_X$ was justified by the fact that CO is treated as an inert pollutant in the model used (see Sect. 2.3), thus varying CO emissions do not affect the modelled $NO_X$ concentrations (and vice versa). The sample values were then used…

**-Section 2.3 – More details should be given about the model set-up. What was the spatial resolution of the model? The text says a 'high resolution' model was used but this is vague. Was the resolution the same as that of the base emissions – 3 km x 3 km?**

We thank the reviewer for this comment. Here the "high resolution" was referred to the possible output resolution of ADMS-Urban, which is not limited by the resolution and format of the input emissions. This is because the model is based on the analytical Gaussian plume formula (and its derivations). If a large number of point locations were defined, the model can produce fine concentration contours (e.g. Biggart et al., 2020) based on the exact distance of these points to emission sources. When outputting in a gridded format, the output resolution can also be finer than that of the gridded emissions. In this case, we only used ADMS-Urban to output pollutant concentrations at measurement locations, and thus did not mention an output resolution.

**-I am led to believe that the model is run for the whole year of 2016 but this isn't clearly stated anywhere when describing the model set-up. Please add some text to make this clearer.**

This is correct and now clarified in the last paragraph under Section 2.3:

The input meteorology data and background pollutant concentrations described above provided the same lateral boundary conditions for the 140 adjusted PEE simulations, among which only the emissions of $NO_X$ (and $NO_2$) varied. An additional simulation forced with these boundary conditions and the base emissions was also performed and is hereinafter referred to as the base run. All 141 simulations were run for the whole year of 2016 to produce hourly pollutant concentrations at each measurement location (see Fig. 1). Output of these simulations were then compared to measurements to derive *a posteriori*

**-Figs 4 and 5. I recommend that the authors add a 'site type' label next to each group of names. I appreciate that the colour coding is described in the figure caption but a label would make it easier for a reader to interpret the figure.**

We thank the reviewer for this suggestion. Labels for site type are added to the site names in both figures:

[Figure]

Figure 4. Mean square errors (MSE) in (a) hourly $NO_2$ concentrations and (b) daily maximum 8-hour mean (MDA8) $O_3$ concentrations associated with the adjusted perturbed emissions ensemble (PEE) simulations, arranged in ascending order of the input annual total $NO_X$ emissions (from left to right), and the base run (marked by black frames) at each long-term monitoring site. In each panel, the MSEs are grouped into quartiles and colour-coded accordingly. The monitoring sites are arranged and colour-coded according to the site type: urban site (magenta), traffic monitoring site (purple), suburban site (orange), clean site (light green) and regional background site (green).

[Figure]

Figure 5. Error component with the highest contribution to the mean square errors (MSE) in (a) hourly NO$_2$ concentrations and (b) daily maximum 8-hour mean (MDA8) O$_3$ concentrations associated with the adjusted perturbed emissions ensemble (PEE) simulations, arranged in ascending order of the input annual total NO$_X$ emissions (from left to right), and the base run (marked by black frames) at each long-term monitoring site. The monitoring sites are arranged and colour-coded according to the site type: urban site (magenta), traffic monitoring site (purple), suburban site (orange), clean site (light green) and regional background site (green).

**- There is no discussion about any seasonal variation in the agreement between the model and the base emissions which would be interesting to see in the results and discussion. Was this investigated and if so, could some details be added?**

We thank the reviewer for this comment. We investigated the interannual variations in the performance of the base run (input with the base emissions) and found larger biases in the mean $NO_2$ and $O_3$ concentrations in the summer months at most urban and traffic monitoring sites, while the biases at suburban, clean and regional background sites showed less seasonal variations. We have attached a figure below. As our conclusions were drawn with respect to annual emissions within the modelling domain, we think the evaluation of the simulations' performance in hourly concentrations presented in the manuscript is appropriate. Nonetheless, we acknowledge that biases in the interannual variations of emissions may be present in the current *a posteriori* estimates.

[Figure]

Figure R2. Biases in the monthly mean NO₂ and O₃ concentrations in 2016 modelled by the base run at each long-term monitoring site.

**-Table 1- Footnote says that nighttime fraction is given as a percentage but I think the table gives it as a ratio.**

We thank the reviewer for pointing out this error. The definitions for the night-time fraction of transport sector NOₓ emissions in the two PEEs are included in a footnote, and the respective

minimum and maximum values adopted are corrected to read (note that the "optimised PEE" is rephrased as the "adjusted PEE" according to suggestion by Referee #1):

Table 1. Emission parameters[a] and the respective uncertainty ranges sampled by the initial and the  adjusted perturbed emissions ensembles (PEEs).

| Parameter | Initial PEE | | Optimised Adjusted PEE | |
|---|---|---|---|---|
| | Min | Max | Min | Max |
| Industry sector ground level $NO_X$ emissions | 0.4 | 1.6 | 0.05 | 1.6 |
| Industry sector elevated $NO_X$ emissions | 0.4 | 1.4 | 0.05 | 1.4 |
| Power sector $NO_X$ emissions[b] | 0.2 | 1.4 | NA | NA |
| Power sector $NO_X$ emissions below 152 m | NA | NA | 0.05 | 1.6 |
| Power sector $NO_X$ emissions above 152 m | NA | NA | 0.05 | 1.6 |
| Residential sector $NO_X$ emissions | 0.4 | 1.5 | 0.05 | 1.5 |
| Transport sector $NO_X$ emissions | 0.4 | 2 | 0.05 | 1.5 |
| Night-time fraction of transport sector $NO_X$ emissions[c] |  10 |  40 |  10 |  30 |

[a] All parameters are defined as ratios of the 2016 emissions to the base emissions from 2013, except for the night-time fraction of transport sector $NO_X$ emissions which is defined as a percentage (%) of the daily totals in 2016.

[b] Power sector $NO_X$ emissions are effectively represented by one parameter in the initial PEE. In the  adjusted PEE, the emissions are split into two parameters, namely emissions below and above 152 m.

[c] Night-time fraction of transport sector $NO_X$ emissions is defined as those occurring during 11am-6am in the initial PEE. In the adjusted PEE, it is modified to $NO_X$ emitted during 0-5 am from the transport sector.

**-Line 422 – in-text description of Fig. 7 before describing Fig. 7(f) would improve readability.**
The start of the paragraph is updated as follows:

On account of the analysis above, we only used observations of $NO_2$ to constrain $NO_X$ emissions, which is also in line with numerous top-down emission optimisation studies using satellite observations of column $NO_2$ (e.g. Lamsal et al. 2011; Martin et al. 2003; Napelenok et al. 2008; Qu et al. 2017). Figure 7 shows the average performance for hourly $NO_2$ of individual PEE simulations against the value set for each emission parameter in Table 1. Figure 7(f)  reveals a strong positive correlation between the median MSE in hourly $NO_2$ of a simulation and the  input transport sector $NO_X$ emissions

**-Line 496 – add 'were' to sentence ...$NO_2$ concentrations were in much...**
**-Line 497 - remove 'were' after observations**
This sentence is amended to read:

[revised manuscript text omitted]

---

## Author Response (AR2)

Comments to the author:
Dear Authors,
Thank you for the submission of a revised manuscript and the authors' response. I find the reviewers' questions well answered and the changes made to the manuscript appropriate. Before I accept the manuscript for publication, one technical issue needs to be resolved. The data availability statement in the current manuscript refers to data archives on websites that are not written in English. Since ACP is a European journal that uses English as language, data should be made available in a repository in English. You can find detailed information on the data policy of ACP on https://www.atmospheric-chemistry-and-physics.net/policies/data_policy.html . The site https://www.re3data.org/ may help to find (institutional) repositories where you can store your data. Please make the data that you used in your paper available in a suitable repository and upload a revised manuscript with a correspondingly corrected data availability statement.
Best regards,
Andreas Hofzumahaus

Dear Dr Hofzumahaus,

Thank you for your review and your help in finding a repository for the data used in this work. We have now deposited the data in a repository which is in English and open access. Our manuscript's data availability statement is also updated accordingly.

Kind regards,

Le Yuan on behalf of all co-authors

---

## Author Response (AR3)

**Remarks from the preceding review file validation**

Please ensure that the colour schemes used in your maps and charts allow readers with colour vision deficiencies to correctly interpret your findings. Please check your figures using the Coblis – Color Blindness Simulator (https://www.color-blindness.com/coblis-color-blindness-simulator/) and revise the colour schemes accordingly.

Dear editor,

Thank you for your comment. We have replaced one figure which did not use a colour-blind friendly colour scheme. All other figures are interpretable with colour vision deficiencies.

Kind regards,

Le Yuan on behalf of all co-authors